# Causal Attribution Analysis for Continuous Outcomes

**Shanshan Luo** [1]   **Yixuan Yu** [1]   **Chunchen Liu** [2]   **Feng Xie** [* 1]   **Zhi Geng** [1]

## Abstract

Previous studies have extensively addressed the attribution problem for binary outcome variables. However, in many practical scenarios, the outcome variable is continuous, and simply binarizing it may result in information loss or biased conclusions. To address this issue, we propose a series of posterior causal estimands for retrospectively evaluating multiple correlated causes from a continuous outcome. These estimands include posterior intervention effects, posterior total causal effects, and posterior natural direct effects. Under assumptions of sequential ignorability, monotonicity, and perfect positive rank, we show that the posterior causal estimands of interest are identifiable and present the corresponding identification equations. We also provide a simple but effective estimation procedure and establish asymptotic properties of the proposed estimators. An artificial hypertension example and a real developmental toxicity dataset are employed to illustrate our method.

## 1. Introduction

In social science (VanderWeele, 2012), health risk assessment (Khoury et al., 2004), legal contexts (Sanders et al., 2021), and explainable artificial intelligence (Galhotra et al., 2021), researchers are interested not only in assessing the effects of causes (Rosenbaum & Rubin, 1983; Robins et al., 1994; Angrist et al., 1996; Bang & Robins, 2005; Ding et al., 2011; Zhao et al., 2012; Jiang et al., 2016; Fan Li & Zaslavsky, 2018; Yang et al., 2020), but also in inferring causes from specific outcomes (Pearl, 1995; Dawid, 2000; Tian & Pearl, 2000; Kuroki & Cai, 2011; Dawid & Musio, 2022; Pearl, 2022; Lu et al., 2023; Li et al., 2023). For instance, for hypertensive patients, researchers may retrospectively evaluate whether the development of hypertension was caused by dietary habits, exercise routines, and physical characteristics. Additionally, such questions are also very common in developmental toxicity risk studies, where researchers in clinical trials aim to determine whether abnormal weight loss in pups is caused by potentially toxic agents, organ disease, or other risk factors.

To explain such retrospective studies, researchers need to use counterfactual scenarios that imagine what would have happened if certain conditions experienced previously had been different (Lu et al., 2023). For example, for a patient with high blood pressure who never exercises, it is inferred what his blood pressure would have been like if there had been an intervention to make him exercise regularly. Similarly, for a pup exposed to a toxic reagent and abnormally thin, it is inferred how its body weight would have changed if it had received a placebo instead. Pearl (2000) introduced a three layer causal hierarchy, comprising association, intervention, and counterfactual levels. Retrospective causal analysis, which assesses the causes of observed effects, falls under the third level of this framework. The first two levels primarily involve predicting or evaluating the effects of interventions. In contrast, the third level focuses on determining whether observed outcomes can be attributed to prior interventions or exposures (Pearl, 2015; Dawid et al., 2014; 2015).

While randomized experiments and standard assumptions effectively address the first two levels of causation (Rosenbaum & Rubin, 1983; Pearl, 2014), they are not enough to address the challenges posed by the third level (Dawid & Musio, 2022). This limitation presents a significant challenge for traditional causal inference methods when dealing with such retrospective analysis problems. To formally answer such questions, Pearl (1999) outlined three counterfactual definitions of causal relationships to capture the necessity or sufficiency of a cause for a given binary effect. Additionally, Dawid et al. (2014) introduced the probability of causation to infer the cause for a given binary effect. When there are multiple potentially correlated causes, Lu et al. (2023) and Li et al. (2023) introduced posterior causal effects under observed post-treatment variables to retrospectively deduce causes from a single effect and multiple effects, respectively. In many clinical trials, the outcome variables of interest may be continuous, such as weight,

[1]School of Mathematics and Statistics, Beijing Technology and Business University, Fangshan District, Beijing, China [2]LingYang Co.Ltd, Alibaba Group, Hangzhou, China. Correspondence to: Feng Xie <fengxie@btbu.edu.cn>.

*Proceedings of the 42nd International Conference on Machine Learning*, Vancouver, Canada. PMLR 267, 2025. Copyright 2025 by the author(s).

blood pressure, and income. However, most existing literature primarily conducts attribution analysis for binary outcomes (Pearl, 1999; Dawid et al., 2014; Lu et al., 2023). Research on continuous outcome variables remains limited, and the formal definitions, identification expressions, and estimation procedures for continuous outcomes require further exploration.

In this paper, we propose a new framework for causal attribution with continuous outcomes. Unlike most prior work which focuses on binary responses, our method is tailored to continuous settings and enables retrospective analysis of how multiple causes contribute to an observed outcome. Our contributions are as follows: First, we extend the posterior attribution framework to continuous outcomes, which introduces new technical challenges and practical relevance, as binarizing outcomes often leads to information loss or bias. Second, under sequential ignorability and the perfect positive rank assumption (Heckman et al., 1997), we prove that individual treatment effects and posterior intervention effects are identifiable. Third, assuming monotonicity among multiple causes (Lu et al., 2023; Li et al., 2023), we establish the identifiability of all proposed estimands and provide explicit identification formulas. Fourth, we develop a novel two-step estimation procedure: we recover individual-level counterfactual mappings and treatment effects, and then estimate the remaining posterior causal estimands based on these mappings. Fifth, we also present simplified identification results under a known directed acyclic graph (DAG), and illustrate the application of our methods using an artificial hypertension example. Proofs of all theoretical results are provided in the Supplementary Materials.

This paper is structured as follows. Section 2 presents the notation and definitions. Section 3 discusses the identifiability of the proposed posterior causal estimands. Section 4 outlines a two-step estimation method for the proposed estimands. In Section 5, we employ an artificial hypertension example to illustrate our proposed method. Finally, Section 6 concludes with a brief summary.

## 2. Notation and definitions

Assuming that we observe $n$ independent and identically distributed samples from a superpopulation. We first consider the scenario with a single cause and a single outcome. For each unit $i$, let $X_i$ represent a binary potential cause, where $X_i = 1$ indicates receiving treatment, and $X_i = 0$ indicates receiving control. Let $Y_i$ be the observed continuous outcome. Let $Y_{i,X=0}$ and $Y_{i,X=1}$ denote the potential outcomes corresponding to $X_i = 0$ and $X_i = 1$, respectively. Many common measurements, such as weight, blood pressure, and income, are typically continuous, but $Y_i$ may fall within a specific interval of interest denoted as $\mathcal{E}_i$. Therefore, given the evidence $(X_i, \mathcal{E}_i)$, we aim to evaluate the effect

of changes in $X_i$ on the outcome $Y_i$, thereby evaluating the likelihood of $X_i$ being the cause of event $\mathcal{E}_i$.

Next, we consider the case with multiple causes $X = (X_1, \ldots, X_p)$ and a single outcome $Y$, where $X$ is a binary vector of causes, and the causes may affect each other. Without loss of generality, we assume that the causes are arranged in a topological order such that $X_l$ is not a cause of $X_k$ for $k < l$. For example, $X$ is a sequence of observations ordered in time, or $X$ consists of variables in a directed acyclic graph (DAG) where $X_k$ is not a descendant of $X_l$ for $k < l$. For generic sets of variables $W$ and $U$, we use $W_u$ to denote the potential outcome of $W$ that would have resulted if $U$ were intervened to level $u$. In particular, if $W = (W_1, \ldots, W_s)$, then $W_u = \{(W_1)_u \ldots, (W_s)_u\}$. We make the consistency assumption that connects observed variables to potential outcomes, i.e., $W_u = W$ if $U = u$ (Pearl, 2015). We further suppose the composition assumption holds in the sense that for any variable sets $W, V$ and $U$, $W_{vu} = W_v$ if $U_v = u$ (Pearl, 2015). The consistency assumption can be viewed as a special case of the composition assumption if $V$ is empty.

To measure how likely $X_k$ is a cause of the continuous effect given observed evidence $(x, \mathcal{E})$, we extend the concept of *posterior total causal effect* (postTCE) defined by Lu et al. (2023) as follows:

$$\begin{aligned} &\text{PostTCE}\left(X_k \Rightarrow Y \mid x, \mathcal{E}\right) \\ &= E\left(Y_{X_k=1} - Y_{X_k=0} \mid x, \mathcal{E}\right), \end{aligned}$$

where we use "$x$" to represent "$X = x$" for notational simplicity. It is important to note that this definition includes the event $\mathcal{E}$ defined by the observed outcome, and cannot simply be considered as a conditional average causal effect (CATE). As advocated by Lu et al. (2023), a larger value of the posterior total causal effect indicates that the effect or outcome is more attributable to the cause $X_k$. The cause that produces the largest posterior total causal effect is usually considered the highest risk factor.

Similar to the direct causal effect considered by Pearl (2000), we define the posterior natural direct effect of $X_k$ on $Y$ given the observed evidence $(x, \mathcal{E})$, which quantifies the effect of $X_k$ on $Y$ not mediated through intermediate variables. Let $A_k = (X_1, \ldots, X_{k-1})$ and $D_k = (X_{k+1}, \ldots, X_p)$. Then $X = (A_k, X_k, D_k)$, and $x = (a_k, x_k, d_k)$ denotes a value of $X$. Given the evidence $(x, \mathcal{E})$, the *posterior natural direct effect* (postNDE) of $X_k$ on $Y$ is:

$$\begin{aligned} &\text{PostNDE}(X_k \Rightarrow Y \mid x, \mathcal{E}) \\ &= E\{Y_{X_k=1, D_k(a_k,0)} - Y_{X_k=0} \mid x, \mathcal{E}\}, \end{aligned}$$

where $D_k(a_k, 0)$ is the potential outcome under $(A_k, X_k) = (a_k, 0)$. Throughout this paper, we use $D_k(a_k, x_k)$ and $(D_k)_{a_k, x_k}$ interchangeably in the nested potential outcomes.

The postNDE describes the effect observed when each individual in the subpopulation $(x, \mathcal{E})$ switches from $X_k = 0$ to $X_k = 1$, while keeping $D_k$ at its value when $X_k = 0$.

Parallel to the natural indirect effect considered by Pearl (2000), we also define the *posterior natural indirect effect* (postNIE) of $X_k$ on $Y$ given the evidence $(x, \mathcal{E})$ as follows:

$$\text{PostNIE}(X_k \Rightarrow Y \mid x, \mathcal{E})$$
$$= E\{Y_{X_k=1} - Y_{X_k=1, D_k(a_k, 0)} \mid x, \mathcal{E}\}.$$

The postNIE quantifies, for each individual in the subpopulation $(x, \mathcal{E})$, the effect observed when $X_k$ is set to $X_k = 1$, while all intermediate variables along the pathway from $X_k$ to $Y$ change from state $D_k(a_k, 1)$ to state $D_k(a_k, 0)$. Through the definitions, we have that:

$$\text{PostTCE}(X_k \Rightarrow Y \mid x, \mathcal{E})$$
$$= \text{PostNDE}(X_k \Rightarrow Y \mid x, \mathcal{E}) + \text{PostNIE}(X_k \Rightarrow Y \mid x, \mathcal{E}).$$

Given the observed evidence $(x, \mathcal{E})$, in addition to assessing the a posteriori total, direct and indirect effects of a particular cause $X_k$, we need to consider assessing the synergistic effects of a joint intervention with all possible causes in an alternative state $X = x'$. Indeed, synergistic effects are important in many applications. For example, having heart disease alone may have a limited effect on blood pressure, whereas the combination of an unhealthy diet and heart disease may jointly contribute to elevated blood pressure. Therefore, the *posterior intervention causal effect* (postICE) for another state $X = x'$ is defined as follows:

$$\text{PostICE}(Y_{x'} \mid x, \mathcal{E}) = E(Y_{x'} - Y \mid x, \mathcal{E}). \quad (1)$$

The *individual treatment effect* (ITE) for any pair $(x', x^*)$ can be defined as $\text{ITE}(x', x^*) = Y_{x'} - Y_{x^*}$, representing the difference in potential outcomes for each individual under two different treatment conditions. Inferring ITEs presents a fundamental challenge because we can only observe one potential outcome for each unit (Rosenbaum & Rubin, 1983).

Table S3 in Section S1 of the Supplementary Material compares our estimands for continuous outcomes with those for binary outcomes in Lu et al. (2023), illustrating their notational alignment and practical relevance.

# 3. Identifiability of posterior causal estimands and required assumptions

## 3.1. Assumptions required for identifiability

Define $W = (X, Y)$ and let $W_{r:s}$ denote a subvector $(W_r, W_{r+1}, \dots, W_s)$ of $W$ for $r \leq s$. Let $w_{r:s}^* = (w_r^*, \dots, w_s^*) \preceq w_{r:s} = (w_r, \dots, w_s)$ denote that $w_i^* \leq$ $w_i$ for all $r \leq i \leq s$. To identify the proposed estimands, we make the following commonly used assumptions in previous studies (Heckman et al., 1997; Pearl, 2000; 2014; Lu et al., 2023; Li et al., 2023).

**Assumption 3.1** (Sequential ignorability). We consider the following assumptions:

(i) there is no confounding between $W_s$ and $W_{1:s-1}$, i.e., $(W_s)_{w_{1:s-1}} \perp\!\!\!\perp W_{1:s-1}$ for all $w_{1:s-1}$ and $s = 2, \dots, p+1$;

(ii) the elements in $\{(W_s)_{w_{1:s-1}}\}_{s=1}^{p+1}$ are mutually independent for any given $w_{1:p}$.

The independence condition in Assumption 3.1 can be relaxed by introducing the baseline covariates, and we omit it for simplicity. The Assumption 3.1(i) implies that the potential outcome of each variable is independent of the prior variables in causal order. Under the Assumption 3.1(i), if $W_s$ has a nonparametric causal structural model $W_s = m_s(W_{1:s-1}, \epsilon_s)$ with an unknown function $m_s(\cdot, \epsilon_s)$ and a error variable $\epsilon_s \perp\!\!\!\perp W_{1:s-1}$, then Assumption 3.1(ii) holds naturally because Assumption 3.1(i) implies that $\epsilon_s \perp\!\!\!\perp \epsilon_{1:s-1}$ for $s = 2, \dots, p+q$, which further implies Assumption 3.1(ii). Assumption 3.1 rules out unobserved confounders between any two variables in $W$. However, each variable $X_k$ may still confound the relationship between $Y$ and $X_l$, or between $X_l$ and $X_s$, provided $k < l, s$. Assumption 3.1 is frequently employed in causal inference with complex systems, including mediation analysis (Imai et al., 2010) and longitudinal data involving time-dependent confounders (Robins, 2000).

**Assumption 3.2** (Monotonicity). For $s = 2, \dots, p$, we assume that $(W_s)_{w_{1:s-1}^*} \leq (W_s)_{w_{1:s-1}}$ whenever $w_{1:s-1}^* \preceq w_{1:s-1}$ holds.

Assumption 3.2 implies that each cause has a non-negative effect on subsequent causes. This assumption is commonly expressed in epidemiology as "no prevention", meaning that no individual is helped by exposure to a risk factor. To identify the posterior causal estimands, Lu et al. (2023) and Li et al. (2023) also introduce the same monotonicity assumption across multiple potentially correlated causes. The validity of monotonicity cannot be tested directly, but under Assumption 3.1, the monotonicity can be falsified by imposing testable restrictions on the observed data distribution. For example, for any $w_{1:s-1}^* \preceq w_{1:s-1}$, the following equality can be used to falsify the monotonicity assumption:

$$\text{pr}(W_s = 1 \mid W_{1:s-1} = w_{1:s-1}^*)$$
$$\leq \text{pr}(W_s = 1 \mid W_{1:s-1} = w_{1:s-1}).$$

**Assumption 3.3** (Perfect positive rank). We assume that $W_{p+1} = m_{p+1}(W_{1:p}, \epsilon_{p+1})$, or equivalently $Y =$

$m_{p+1}(X, \epsilon_{p+1})$, where $\epsilon_{p+1}$ represents a scalar-valued error variable. The unknown link function $m_{p+1}(X, \cdot)$ is continuous and strictly increasing in $\epsilon_{p+1}$.

For continuous outcome variables, Assumption 3.3 in our framework states that the individual-level outcome is a monotonic function of an unobserved latent variable $\epsilon_{p+1}$. This implies that individuals preserve their relative ranks across different treatment conditions. Such rank-preserving behavior is commonly used to identify individual treatment effects or quantile treatment effects in counterfactual causal inference literature (Heckman et al., 1997; Chernozhukov & Hansen, 2005), but it is a relatively novel assumption in retrospective attribution analysis. For the linear model,

$$Y = \alpha_0 + \alpha_1 X_1 + \cdots + \alpha_p X_p + \epsilon_{p+1}, \qquad (2)$$

Assumption 3.3 holds naturally. Note that Assumption 3.3 imposes a condition only on the outcome variable $X_{p+1} = Y$, while Assumption 3.2 concerns the structural relationship among multiple causes $(X_1, \ldots, X_p)$. Since these two assumptions relate to different variables, the former does not imply the latter.

Lu et al. (2023) considered the monotonicity assumption of the binary outcome with respect to multiple correlated causes $X$, denoted as $Y_{X=x^*} \leqslant Y_{X=x}$ for any $x^* \preceq x$. However, this monotonic relationship may not be applicable when the outcome variable is continuous. Specifically, the condition implies that increasing any component of the treatment vector, while keeping others fixed or increased, should not decrease the outcome. In the linear model (2), satisfying this monotonicity assumption requires all coefficients to be non-negative, i.e., $\alpha_j \geq 0$ for all $j = 1, \ldots, p$, which may be overly restrictive for continuous outcomes.

The basic restriction in Assumption 3.3 is also referred to as the rank preservation or rank invariance (Heckman et al., 1997; Chernozhukov & Hansen, 2005; Vuong & Xu, 2017; Feng et al., 2020). If an individual with regular exercise and without heart disease (i.e., $x = (1, 0)$) has the lowest blood pressure in the subpopulation $\{(X_i, Y_i) : X_i = x\}$, then according to Assumption 3.3, an individual with no exercise and heart disease (i.e., $x' = (0, 1)$) should also have the lowest blood pressure in the corresponding subpopulation $\{(X_i, Y_i) : X_i = x'\}$; and vice versa. The strict monotonic increase of $\epsilon$ can be changed to the strict monotonic decrease without affecting the subsequent discussion. For simplicity, we assume the strict monotonic increase. For any given error $\epsilon_{p+1}^*$, Assumption 3.3 requires that the relative rank or quantile of $Y_x \equiv m_{p+1}(x, \epsilon_{p+1}^*)$ be the same as that of $Y_{x'} \equiv m_{p+1}(x', \epsilon_{p+1}^*)$ for any $x \neq x'$. A stronger version of Assumption 3.3 assumes that the error term $\epsilon_{p+1}$ is additive, that is, $Y = m_{p+1}^*(X) + \epsilon_{p+1}$ for some real-valued function $m_{p+1}^*(\cdot)$. Moreover, the heteroscedasticity model also satisfies Assumption 3.3: $Y = m_{p+1}^*(X) + \sigma(X)\epsilon_{p+1}$,

for some real-valued function $m_{p+1}^*(X)$ and positive function $\sigma(X)$.

### 3.2. Identification equations of posterior causal estimands

Under Assumptions 3.1 and 3.3, we first consider the identifiability of ITEs and postICEs. Let $\mathcal{S}_{Y_x}$ denote the support of the potential outcome $Y_x$, which can be identified by $\mathcal{S}_{Y|X=x}$ under Assumption 3.1. The key to our identification strategy is to match the potential outcome $Y_x \equiv m_{p+1}(x, \epsilon_{p+1})$ with another potential outcome $Y_{x'} \equiv m_{p+1}(x', \epsilon_{p+1})$ through a mapping $\phi_{x \to x'}(\cdot)$, such that $Y_{x'} = \phi_{x \to x'}(Y_x)$. This mapping $\phi_{x \to x'}(\cdot)$ is termed a counterfactual mapping (Vuong & Xu, 2017; Feng et al., 2020), because it allows us to find the counterfactual outcome $Y_{x'}$ from $Y_x$ using the function $\phi_{x \to x'}(\cdot)$, and vice versa.

Let $m_{p+1}^{-1}(x, \cdot)$ be the inverse function of $m_{p+1}(x, \cdot)$. According to Assumption 3.3, for any pair $(X, Y) = (x, y)$, we can uniquely represent the error term as $\epsilon_{p+1} = m_{p+1}^{-1}(x, y)$, although the specific form of $m_{p+1}^{-1}(x, y)$ is unknown. Therefore, $Y_{x'}$ is uniquely defined by $\phi_{x \to x'}(y) \equiv m_{p+1}\{x', m_{p+1}^{-1}(x, y)\}$ for each $y \in \mathcal{S}_{Y_x}$. Moreover, the counterfactual mapping $\phi_{x \to x'}(\cdot)$ is a continuous and strictly increasing function from $\mathcal{S}_{Y_x}$ onto $\mathcal{S}_{Y_{x'}}$. Thus, if we can identify $\phi_{x \to x'}(y)$ for all $y \in \mathcal{S}_{Y_x}$ and $x \neq x'$, we can recover ITEs for each individual.

**Lemma 3.4.** *Under Assumptions 3.1 and 3.3, for any $y \in \mathcal{S}_{Y_x}$, the counterfactual mapping $\phi_{x \to x'}(\cdot)$ is identified by the continuous extension of*

$$\phi_{x \to x'}(y) = F_{x'}^{-1}\{F_x(y)\}, \quad \forall y \in \mathcal{S}_{Y_x}^{\circ}, \qquad (3)$$

*where $F_x(y) = \mathrm{pr}(Y \leq y \mid X = x)$ and $\mathcal{S}_{Y_x}^{\circ}$ is the interior of $\mathcal{S}_{Y_x}$. Moreover, the ITEs of every individual in the population can be identified.*

Lemma 3.4 establishes the identifiability of the counterfactual mapping $\phi_{x \to x'}(\cdot)$ on $\mathcal{S}_{Y_x}$ constructively by matching the quantiles of $Y_x$ and $Y_{x'}$. We provide further intuition for Lemma 3.4. When we observe an individual in the subpopulation $\{(X_i = x, Y_i)\}$ with the highest blood pressure, we can recover the joint distribution of individual $i$ by identifying the individual with the highest blood pressure in each observed subgroup $\{(X_j = x', Y_j)\}$ for any $x' \neq x$. Given the identifiability of the counterfactual mapping and ITEs, the postICEs can also be identified using an inverse probability weighting expression (Horvitz & Thompson, 1952),

$$\mathrm{PostICE}\,(Y_{x'} \mid x, \mathcal{E}) = E\left[\frac{\mathbb{I}(X = x, \mathcal{E})}{\mathrm{pr}(X = x, \mathcal{E})}\{\phi_{x \to x'}(Y) - Y\}\right], \tag{4}$$

where $\mathbb{I}(\cdot)$ denotes the indicator function.

Assumptions 3.1 and 3.3 establish the identifiability of ITEs and postICEs. However, they are not sufficient to ensure the identifiability of other posterior causal estimands (e.g., postNDEs and PostNIEs) when considering a specific cause $X_k$. This is due to the challenge posed by identifying expectations of the nested potential outcomes given the observed evidence. For example, the postNDE involves the conditional expectation of the cross-world intervention potential outcome $Y_{X_k=1,D_k(a_k,0)}$, which cannot be identified using Lemma 3.4 alone. Before formally identifying these expectations, we provide another lemma for identifying the conditional probability of the counterfactual outcomes of the causes.

**Lemma 3.5.** *Under Assumptions 3.1 and 3.2, given the observed evidence $(a_k, x_k, d_k, \mathcal{E})$, let $d_k^* = (x_{k+1}^*, \ldots, x_p^*)$ and $d_k = (x_{k+1}, \ldots, x_p)$.*

*(i) For $d_k^* \preceq d_k$, we have,*

$$\text{pr}\{D_k(a_k, 0) = d_k^* \mid a_k, 1, d_k\}$$
$$= \prod_{s=k+1}^{p}\{(1 - x_s^*) + (2x_s^* - 1)x_s R_{0s}\},$$

*where $R_{0s} = \dfrac{\text{pr}(X_s = 1 \mid a_k, 0, x_{k+1}^*, \ldots, x_{s-1}^*)}{\text{pr}(X_s = 1 \mid a_k, 1, x_{k+1}, \ldots, x_{s-1})}$.*

*(ii) For $d_k \preceq d_k^*$, we have,*

$$\text{pr}\{D_k(a_k, 1) = d_k^* \mid a_k, 0, d_k\}$$
$$= \prod_{s=k+1}^{p}\{x_s^* + (1 - 2x_s^*)(1 - x_s)R_{1s}\},$$

*where $R_{1s} = \dfrac{\text{pr}(X_s = 0 \mid a_k, 1, x_{k+1}^*, \ldots, x_{s-1}^*)}{\text{pr}(X_s = 0 \mid a_k, 0, x_{k+1}, \ldots, x_{s-1})}$.*

**Theorem 3.6.** *Under Assumptions 3.1-3.3, given the evidence $(a_k, x_k, d_k, \mathcal{E})$, the postNDE, postNIE, and postTCE of $X_k$ on $Y$ can be identified as follows:*

*(i) given $x_k = 1$, for any $x_k^\star \in \{0, 1\}$, we have,*

$$E\{Y_{x_k^\star, D_k(a_k, 1)} \mid x, \mathcal{E}\} = E(Y_{x_k^\star, d_k} \mid x, \mathcal{E}),$$
$$E\{Y_{x_k^\star, D_k(a_k, 0)} \mid x, \mathcal{E}\}$$
$$= \sum_{d_k^* \preceq d_k} E(Y_{x_k^\star, d_k^*} \mid x, \mathcal{E})\text{pr}\{D_k(a_k, 0) = d_k^* \mid x\},$$

*(ii) given $x_k = 0$, for any $x_k^\star \in \{0, 1\}$, we have,*

$$E\{Y_{x_k^\star, D_k(a_k, 0)} \mid x, \mathcal{E}\} = E(Y_{x_k^\star, d_k} \mid x, \mathcal{E}),$$
$$E\{Y_{x_k^\star, D_k(a_k, 1)} \mid x, \mathcal{E}\}$$
$$= \sum_{d_k \preceq d_k^*} E(Y_{x_k^\star, d_k^*} \mid x, \mathcal{E})\text{pr}\{D_k(a_k, 1) = d_k^* \mid x\},$$

*where $E(Y_{x_k^\star, d_k^*} \mid x, \mathcal{E})$ can be identified by Lemma 3.4, and $\text{pr}\{D_k(a_k, x_k') = d_k^* \mid x\}$ for $x_k' \in \{0, 1\}$ can be identified by Lemma 3.5.*

Theorem 3.6 illustrates that with the additional monotonicity assumption 3.2, we can also identify postNDE, postNIE, and postTCE. In some cases, we may only want to conduct attribution analysis for continuous effects based on evidence from a subset of $(X, \mathcal{E})$, while Lemma 3.4 and Theorem 3.6 are based on fully observed evidence $(x, \mathcal{E})$. For some $s < p$, let $(X', \mathcal{E}) = (X_{i_1}, \ldots, X_{i_s}, \mathcal{E})$ denote a subset of $(X, \mathcal{E})$. We can summarize the remaining set $X \setminus X'$ to obtain the expected results for the subset $(X', \mathcal{E})$ based on Lemma 3.4 and Theorem 3.6. To simplify the exposition, we omit this part. We also refer to Corollary 1 in Lu et al. (2023) and Theorem 3 in Li et al. (2023) for parallel results.

### 3.3. Identification equations of posterior causal effects under causal networks

In this section, we consider the causal structure of observed variables $(X, Y)$ represented by a directed acyclic graph (DAG) or network. We aim to present the simplified identification expressions for the posterior causal estimands given a known DAG. For $k = 1, \ldots, p$, let $\text{Pa}_k$ and $\text{Pa}_Y$ denote the sets of parents of $X_k$ and $Y$ in the graph, respectively. The joint probability distribution $\text{pr}(X, Y)$ can be factorized as $\text{pr}(X, Y) = \prod_{k=1}^{p} \text{pr}(X_k \mid \text{Pa}_k) \text{pr}(Y \mid \text{Pa}_Y)$. We assume $\text{pr}(x, y) > 0$ for each $(x, y)$. For a given graph, the sequential ignorability assumption (Assumption 3.1) posits that there is no unobserved variable intervening between any two or more nodes in the DAG. The monotonicity assumption 3.2 implies that each node $X_k$ has a positive individual monotonic effect on its child nodes. Furthermore, Assumption 3.3 can be simplified such that the unknown link function $m_{p+1}(\text{Pa}_Y, \epsilon_{p+1})$ only needs to be continuous with respect to $\text{Pa}_Y$ and strictly increasing in $\epsilon_{p+1}$. Specifically, the continuous outcome variable is only required to perfectly match the quantiles of potential outcomes under different realizations of its parent nodes $\text{Pa}_Y$. We define a simpler counterfactual mapping $\phi_{\text{pa}_y \to \text{pa}_y'}(\cdot)$ to characterize the mapping relationship of the parent nodes of $Y$ under different realizations $\text{pa}_y$ and $\text{pa}_y'$. The theoretic results presented in Lemma 3.4 and (4) can be simplified for a given graph.

**Corollary 3.7.** *Suppose that the causal network of $(X_1, \ldots, X_p, Y)$ is a DAG. Then, under Assumptions 3.1 and 3.3, for any $x \neq x'$ and $y \in \mathcal{S}_{Y_x}$, let $\text{pa}_y \subset x$ and $\text{pa}_y' \subset x'$, the counterfactual mapping $\phi_{x \to x'}(\cdot)$ can be reduced as follows:*

$$\phi_{x \to x'}(y) = \phi_{\text{pa}_y \to \text{pa}_y'}(y) = F_{\text{pa}_y'}^{-1}\{F_{\text{pa}_y}(y)\}, \quad \forall y \in \mathcal{S}_{Y_x}^\circ,$$

*where $m_{\text{pa}_y}(y) = \text{pr}(Y \leq y \mid \text{Pa}_Y = \text{pa}_y)$. Moreover, the ITEs of every individual in the population can be identified.*

*The following equation also holds:*

$$E\left(Y_{x'} \mid x, \mathcal{E}\right) = E(Y_{\mathrm{pa}_y'} \mid \mathrm{pa}_y, \mathcal{E})$$
$$= E\left\{\frac{\mathbb{I}(\mathrm{Pa}_y = \mathrm{pa}_y, \mathcal{E})}{\mathrm{pr}(\mathrm{Pa}_y = \mathrm{pa}_y, \mathcal{E})}\phi_{\mathrm{pa}_y \to \mathrm{pa}_y'}(Y)\right\},$$

*and the postICE is identifiable.*

Corollary 3.7 indicates that the counterfactual mapping depends only on $\mathrm{Pa}_Y$, the postICEs and ITEs can be calculated using the low-dimensional probabilities $\mathrm{pr}(Y \leq y \mid \mathrm{pa}_y)$. Similarly, the identification equations of the conditional probability $\mathrm{pr}\{D_k(a_k, 0) = d_k^* \mid x\}$ can also be simplified by replacing $R_{0s}$ and $R_{1s}$ in Lemma 3.5 with $R_{0s}^*$ and $R_{1s}^*$:

$$R_{0s}^* = \mathrm{pr}(X_s = 0 \mid \mathrm{pa}_s^*)/\mathrm{pr}(X_s = 0 \mid \mathrm{pa}_s),$$
$$R_{1s}^* = \mathrm{pr}(X_s = 1 \mid \mathrm{pa}_s^*)/\mathrm{pr}(X_s = 1 \mid \mathrm{pa}_s), \qquad (5)$$

where $\mathrm{pa}_s^* \subset (a_k, 0, x_{k+1}^*, \ldots, x_{s-1}^*)$ and $\mathrm{pa}_s \subset (a_k, 1, x_{k+1}, \ldots, x_{s-1})$. The following Corollary 3.8 provides a simplified identification equation of other posterior causal estimands for a given graph.

**Corollary 3.8.** *Suppose that the causal network of $(X_1, \ldots, X_p, Y)$ is a DAG. Under Assumptions 3.1-3.3, given the observed evidence $(a_k, x_k, d_k, \mathcal{E})$, the postNDE, postNIE, and postTCE of $X_k$ on $Y$ can be identified using the following equations:*

*(i) when $x_k = 1$, for any $x_k^\star \in \{0, 1\}$, we have,*

$$E\{Y_{x_k^\star, D_k(a_k, 1)} \mid x, \mathcal{E}\} = E(Y_{\mathrm{pa}^\star} \mid \mathrm{pa}_y, \mathcal{E}),$$
$$E\{Y_{x_k^\star, D_k(a_k, 0)} \mid x, \mathcal{E}\} =$$
$$\sum_{d_k^* \preceq d_k} E(Y_{\mathrm{pa}_y^*} \mid \mathrm{pa}_y, \mathcal{E})\mathrm{pr}\{D_k(a_k, 0) = d_k^* \mid x\},$$

*(ii) when $x_k = 0$, for any $x_k^\star \in \{0, 1\}$, we have,*

$$E\{Y_{x_k^\star, D_k(a_k, 0)} \mid x, \mathcal{E}\} = E(Y_{\mathrm{pa}^\star} \mid \mathrm{pa}_y, \mathcal{E}),$$
$$E\{Y_{x_k^\star, D_k(a_k, 1)} \mid x, \mathcal{E}\} =$$
$$\sum_{d_k \preceq d_k^*} E(Y_{\mathrm{pa}_y^*} \mid \mathrm{pa}_y, \mathcal{E})\mathrm{pr}\{D_k(a_k, 1) = d_k^* \mid x\},$$

*where $E(Y_{\mathrm{pa}_y^\star} \mid \mathrm{pa}_y, \mathcal{E})$ and $E(Y_{\mathrm{pa}_y^*} \mid \mathrm{pa}_y, \mathcal{E})$ can be identified by Corollary 3.7 for $\mathrm{pa}_y^\star \subset (a_k, x_k^\star, d_k)$ and $\mathrm{pa}_y^* \subset (a_k, x_k^\star, d_k^*)$, and $\mathrm{pr}\{D_k(a_k, x_k') = d_k^* \mid x\}$ can be identified by Lemma 3.5 and (5) for any $x_k' \in \{0, 1\}$.*

The theoretic results in this section are very useful in practice and can greatly simplify the computation, especially when the dimensionality of the causes is large. We suggest that practitioners first obtain a simple DAG, or a larger one but with the correct order of nodes, from expert knowledge or conditional independence tests.

## 4. Estimation

In this section, we propose a simple but effective method for estimating the counterfactual outcome mapping $\phi_{x \to x'}(\cdot)$ as well as the posterior causal estimands. Let $\{(X_i, Y_i) : i = 1, \cdots, n\}$ be the independent and identically distributed samples generated according to Assumptions 3.1-3.3. Our estimation procedure consists of two steps: first, for each observation $(X_i, Y_i) = (x, y)$, we estimate the counterfactual mapping $\phi_{x \to x'}(y)$ for $x' \neq x$ using a simple estimator that minimizes a convex population objective function and constructs pseudo samples of the counterfactual outcomes for all individuals. In the second step, we nonparametrically estimate the posterior causal estimands based on the counterfactual mapping $\phi_{x \to x'}(\cdot)$. For simplicity, let $\mathcal{S}_{Y_x}$ denote a compact interval $[y_x^l, y_x^u]$ for any $X = x$, where $-\infty < y_x^l < y_x^u < +\infty$. We also assume the compact support $[y_x^l, y_x^u]$ is known. Otherwise, it can be estimated using the methods proposed in Korostelev & Tsybakov (2012). To establish the asymptotic properties of the estimators to be proposed in this section, we make the following assumptions.

**Assumption 4.1.** (i) The function $m_{p+1}(x, \epsilon_{p+1})$ is continuously differentiable in error term $\epsilon_{p+1}$ for any $X = x$; (ii) The probability density function of the error term $\epsilon_{p+1}$ is continuous; (iii) $\inf_{y \in [y_x^l, y_x^u]} g_x(y) > 0$ for any $X = x$, where $g_x(y) = \partial F_x(y)/\partial y$.

Assumptions 4.1(i) and 4.1(ii) are regularity conditions and, together with Assumptions 3.1 and 3.3, ensure that the marginal distribution $F_x(y)$ is absolutely continuous with respect to the Lebesgue measure, and its probability density function $g_x(y)$ is also continuous for any $X = x$. Assumption 4.1(iii) is introduced for the sake of simplicity in explanation. Trimming techniques can be employed to relax this assumption, but they may introduce technical complexities.

We now aim to derive a counterfactual outcome mapping $\phi_{x \to x'}(\cdot)$ from two marginal distributions $F_x(\cdot)$ and $F_{x'}(\cdot)$. For a given $y \in \mathbb{R}$, we define the objective function as follows: $\rho_{x \to x'}(t; y) = E\{\mathrm{sign}(Y - y) \mid X = x\} \times t - E(|Y - t| \mid X = x')$, where $\mathrm{sign}(u) \equiv 2 \times \mathbb{I}(u > 0) - 1$. The above objective function is motivated by the quantile regression method in Koenker & Bassett (1978) and Feng et al. (2020). Simple calculations reveal that the first-order and second-order conditions of this objective function are given by:

$$\frac{\partial \rho_{x \to x'}(t; y)}{\partial t} = 2\{F_{x'}(t) - F_x(y)\} = 0,$$
$$\frac{\partial^2 \rho_{x \to x'}(t; y)}{\partial t^2} = 2g_{x'}(t) \geq 0.$$

The following lemma demonstrates that for any $y \in \mathcal{S}_{Y_x}^\circ$, the objective function $\rho_{x \to x'}(\cdot; y)$ is uniquely minimized at

the counterfactual outcome $\phi_{x \to x'}(y)$.

**Lemma 4.2.** *Under Assumptions 3.1-3.3, $\rho_{x \to x'}(\cdot; y)$ is continuously differentiable and weakly convex on $\mathbb{R}$. Additionally, $\rho_{x \to x'}(\cdot; y)$ is strictly convex on $\mathcal{S}_{Y_{x'}}^\circ$, and uniquely minimized on $\mathbb{R}$ at $\phi_{x \to x'}(y)$ when $y \in \mathcal{S}_{Y_x}^\circ$.*

The objective function $\rho_{x \to x'}(\cdot; y)$ in Lemma 4.2 has several important properties that support the validity of our estimation procedure. It is continuously differentiable, which ensures that gradient-based optimization methods can be effectively applied. Moreover, the function $\rho_{x \to x'}(\cdot; y)$ is strictly convex in the interior of the support, which implies the uniqueness of the solution. Lemma 4.2 also provides the foundation for our nonparametric estimation of the counterfactual mappings and posterior causal estimands. For the $i$ th observational unit $(X_i = x, Y_i)$, let

$$\hat{\rho}_{x \to x'}(t; Y_i) = \frac{\sum_{j=1}^n \text{sign}(Y_j - Y_i) \times \mathbb{I}(X_j = x)}{\sum_{j=1}^n \mathbb{I}(X_j = x)} \times t$$
$$- \frac{\sum_{j=1}^n |Y_j - t| \times \mathbb{I}(X_j = x')}{\sum_{j=1}^n \mathbb{I}(X_j = x')}.$$

Hence, we can estimate the counterfactual outcome of the $i$ th unit by,

$$\hat{\phi}_{x \to x'}(Y_i) = \underset{t \in [y_{x'}^l, y_{x'}^u]}{\arg\min} \hat{\rho}_{x \to x'}(t; Y_i), \quad \text{if } X_i = x. \quad (6)$$

By separately minimizing $\hat{\rho}_{x \to x'}(\cdot; Y_i)$ for each unit $i$ with $(X_i = x, Y_i)$, we can estimate the counterfactual outcome for an individual $i$ under state $X = x'$ by the counterfactual mapping $\hat{\phi}_{x \to x'}(Y_i)$ in (6), and the individual treatment effect $\text{ITE}(x', x^*)$ can be estimated as follows: $\hat{\Delta}_{\text{ITE},i}(x', x^*) = \hat{\phi}_{x \to x'}(Y_i) - \hat{\phi}_{x \to x^*}(Y_i)$. Moreover, other posterior causal estimands can also be constructed using similar moment estimators.

## 5. Example: risk factors for hypertension

In this section, we will use the example provided in Lu et al. (2023) to explain our proposed causal attribution estimands, and assess the posterior causal effects of risk factors on continuous hypertension. Figure 1 presents the causal network and the corresponding conditional probabilities, where Exercise (E), Diet (D), Heart Disease (HD), Heartburn (Hb), and Chest Pain (CP) are potential causes of hypertension. Let $E = 1$ denote no daily exercise, $D = 1$ denote unhealthy diet, $HD = 1$ denote heart disease, $Hb = 1$ denote heartburn, $CP = 1$ denote chest pain, and BP denote continuous blood pressure $Y$. We generate the data without unobserved confounders to satisfy Assumption 3.1. We also ensure that the relationships among the causes satisfy the monotonicity assumption (Assumption 3.2). For instance, lack of daily exercise ($E = 1$) and poor diet ($D = 1$)

Table 1: Results of posterior intervention causal effects based on different evidence.

| PostICE$(Y_{x'} \mid x, Y > 140)$ | $(x_1, x_4)$ $= (0,0)$ | $(x_1, x_4)$ $= (0,1)$ | $(x_1, x_4)$ $= (1,0)$ | $(x_1, x_4)$ $= (1,1)$ |
|---|---|---|---|---|
| $(x_1', x_4') = (0,0)$ | 0.00 | -12.43 | -1.76 | -18.19 |
| $(x_1', x_4') = (0,1)$ | 3.34 | 0.00 | 2.34 | -5.43 |
| $(x_1', x_4') = (1,0)$ | 1.75 | -10.01 | 0.00 | -15.75 |
| $(x_1', x_4') = (1,1)$ | 12.05 | 5.57 | 10.86 | 0.00 |

are not preventive for heart disease HD. The various potential outcomes also satisfy the perfect rank assumption (Assumption 3.3). Specifically, we modeled blood pressure $Y$ as a continuous outcome variable and ensured that, after binarization, its distribution matches that observed in Lu et al. (2023). The topological order in Figure 1 is $X = (X_1, \ldots, X_5) = (E, D, Hb, HD, CP)$. The joint probability of $X$ and $Y$ is calculated by substituting the conditional probabilities from Figure 1 into the following equation: $\text{pr}(E, D, Hb, HD, CP, BP) = \text{pr}(E) \text{pr}(D) \text{pr}(Hb \mid D) \text{pr}(HD \mid E, D) \text{pr}(CP \mid Hb, HD) \text{pr}(BP \mid HD)$. Without loss of generality, we consider the event $\mathcal{E} = \mathbb{I}(Y > 140)$, which indicates whether hypertension is present. The posterior causal estimands can be used to analyze the causes of hypertension for a patient based on the evidence $(x, Y > 140)$.

We first present the postICEs for hypertension, based on different observed evidence, as shown in Table 1. According to Corollary 3.7, we know that the postICEs are only related to the parent nodes of BP, namely $X_1$ (exercise) and $X_4$ (heart disease). We find that the largest change in postICEs occurs with the evidence $(x_1, x_4, \mathcal{E}) = (1, 1, Y > 140)$ and the intervened treatment $(x_1', x_4') = (0, 0)$. Specifically, for individuals who do not exercise, have heart disease, and have high blood pressure, if they had exercised previously and did not have heart disease, i.e., receiving treatment $(x_1', x_4') = (0, 0)$, their blood pressure would significantly decrease by 18.19, that is, $\text{PostICE}(Y_{x'} \mid X_1 = 1, X_4 = 1, Y > 140) \approx -18.19$. Conversely, in the evidence $(x_1, x_4, \mathcal{E}) = (0, 0, Y > 140)$, i.e., individuals who exercise, do not have heart disease, and have high blood pressure, if they had not exercised previously and did not have heart disease, i.e., receiving $(x_1', x_4') = (1, 0)$, their blood pressure would slightly increase by 1.75, that is, $\text{PostICE}(Y_{x'} \mid X_1 = 0, X_4 = 0, Y > 140) \approx 1.75$.

Since our data generation mechanism ensures the exact same observed data distribution as described in Lu et al. (2023) after binarizing outcome $Y$, we directly use their posterior total causal effects in their Table 1 for comparative analysis. We use the symbol asterisks (*) to differentiate from the definitions provided in this paper. Specifically, let $Y_{X_k = x_k}^*$ be the binary event $\mathbb{I}(Y_{X_k = x_k} > 140)$, we adopt the definitions of the binary outcomes from Lu et al. (2023), denoted as $\text{PN}^*(X_k \Rightarrow \mathcal{E})$ and $\text{postTCE}^*(X_k \Rightarrow \mathcal{E} \mid$

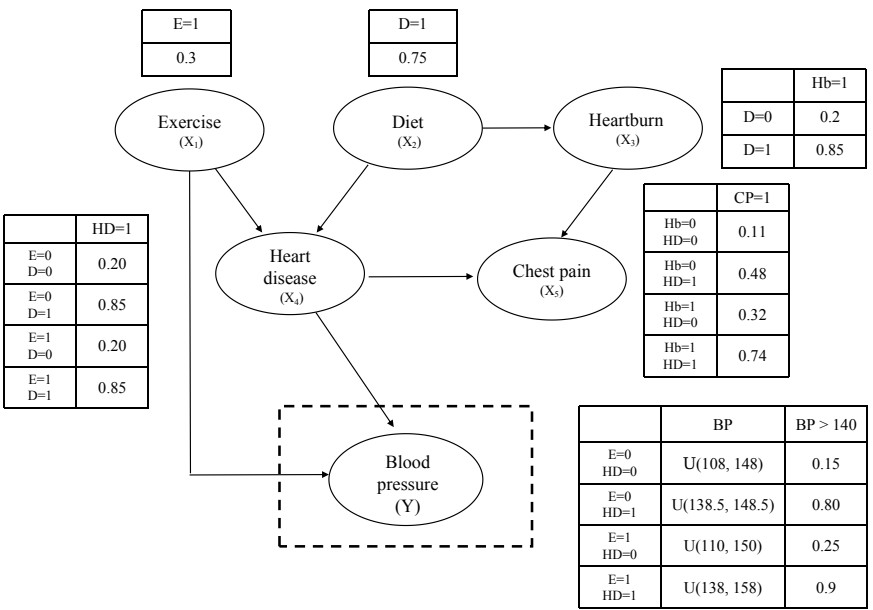

Figure 1: A causal network representing hypertension and its risk factors, where $U(a, b)$ denotes a uniform distribution on the interval $[a, b]$.

Table 2: Results of marginal probabilities of necessity and posterior causal estimands based on the evidence $\{X = (1, 1, 1, 1, 1), Y > 140\}$.

|  | $X_1$ | $X_2$ | $X_3$ | $X_4$ | $X_5$ |
|---|---|---|---|---|---|
| $\text{PN}^*(X_k \Rightarrow \mathcal{E})$ | 0.347 | 0.230 | 0.133 | 0.760 | 0.563 |
| $\text{postTCE}^*(X_k \Rightarrow \mathcal{E} \mid x, Y > 140)$ | 0.344 | 0.207 | 0 | 0.722 | 0 |
| $\text{postNDE}(X_k \Rightarrow Y \mid x, Y > 140)$ | 3.823 | 0 | 0 | 17.023 | 0 |
| $\text{postNIE}(X_k \Rightarrow Y \mid x, Y > 140)$ | 6.805 | 4.561 | 0 | 0 | 0 |
| $\text{postTCE}(X_k \Rightarrow Y \mid x, Y > 140)$ | 10.628 | 4.561 | 0 | 17.023 | 0 |

$x, Y > 140)$, as follows: $\text{PN}^*(X_k \Rightarrow \mathcal{E}) = \text{pr}(Y^*_{X_k=0} = 0 \mid a_k, X_k = 1, d_k, Y > 140)$,

$$\text{postTCE}^*(X_k \Rightarrow \mathcal{E} \mid x, Y > 140)$$
$$= E(Y^*_{X_k=1} - Y^*_{X_k=0} \mid x, Y > 140).$$

The above definitions can be identified using Lemma 1 and Theorem 1 in Lu et al. (2023). We do not consider the causal estimands $\text{postNDE}^*(X_k \Rightarrow \mathcal{E} \mid x, Y > 140)$ and $\text{postNIE}^*(X_k \Rightarrow \mathcal{E} \mid x, Y > 140)$, because there is no literature to support the identifiability results for these two causal quantities after binarization.

Given the observed evidence $\{X = (1, 1, 1, 1, 1), Y > 140\}$, Table 2 presents the results for each potential risk factor $X_k$ with respect to posterior causal estimands. The first row of Table 2 displays the probabilities of necessity $\text{PN}^*(X_k \Rightarrow \mathcal{E})$ computed after binarizing the outcome variable $Y$, while the second row presents $\text{postTCE}^*(X_k \Rightarrow \mathcal{E} \mid x, Y > 140)$. The third to fifth rows present the results for postNDEs, PostNIEs, and PostTCEs considered in this paper. We find that the second and fifth rows of the

table show very similar results in terms of sign and ordering. Specifically, HD, E, and D (denoted as $X_1$, $X_2$, and $X_4$, respectively) all have non-zero postTCE values, indicating that they are risk factors for blood pressure. Among them, HD (i.e., $X_4$) has the largest postTCE, indicating that HD is the most important risk factor for blood pressure. In addition, the value of $\text{PN}^*(X_4 \Rightarrow \mathcal{E})$ is also the largest, further confirming the importance of HD as a risk factor. Notably, for HD (i.e., $X_4$), it can be observed that the postNDE is equal to the postTCE of BP. On the other hand, the postNDE for E (i.e., $X_1$) is smaller than the postTCE for BP, implying that E has direct and indirect causal effects on BP. The third to fifth rows of the table present the postNDE, postNIE, and postTCE values for Hb and CP (denoted as $X_3$ and $X_5$, respectively), which are all equal to zero. This indicates that they are not risk factors for blood pressure. From the causal network in Figure 1, it can be observed that there is no causal path from Hb (i.e., $X_3$) and CP (i.e., $X_4$) to BP. However, $\text{PN}^*(X_5 \Rightarrow \mathcal{E})$ is greater than $\text{PN}^*(X_1 \Rightarrow \mathcal{E})$ and $\text{PN}^*(X_2 \Rightarrow \mathcal{E})$ as shown in the first row.

We also present additional analysis conclusions under different evidence in Section S8 of the Supplementary Material. To assess the stability of the proposed estimation procedure in Section 4 of the Supplementary Material, we conduct simulation studies by generating data according to the causal network depicted in Figure 1. The estimated results of Table 2 under different sample sizes were provided. The simulation results indicate negligible biases and small standard errors, particularly for large sample sizes. For detailed information on data generation procedures and simulation results,

please refer to Section S9 of the Supplementary Material. In Section S10 of the Supplementary Material, we also apply the proposed method to a real dataset (NTP, 2023).

## 6. Discussion

Dawid et al. (2014) pointed out that statistical inference about the causes of effects is particularly challenging and is difficult to justify even in randomized experiments. While some prior research has addressed attribution analysis problems with binary outcomes (Pearl, 2000; Dawid et al., 2015; Dawid & Musio, 2022; Li et al., 2023), evaluating the causes of continuous outcomes or effects remains underexplored. The identifiability of ITEs and postICEs with multiple potentially correlated causes is motivated by the identification of ITEs in the case of a single cause (Heckman et al., 1997). However, even if ITEs are identified, it is insufficient for retrospective assessment of a specific cause of effects, as the monotonicity assumption 3.2 is still necessary. In practice, for estimation purposes, we can begin by recovering the counterfactual mappings between different potential outcomes as well as ITEs through quantile regression (Koenker & Bassett, 1978; Heckman et al., 1997; Feng et al., 2020); then, we can utilize the identification expressions in Lemma 3.4 and Theorem 3.6 for nonparametric estimation.

Our work is also related to recent studies that explain causal attribution using ideas from information theory and Shapley values. Specifically, Schamberg et al. (2020) uses information measures to separate direct and indirect effects, and Jung et al. (2022) suggests a method based on Shapley values to measure the importance of different factors in causal models. These methods mainly focus on average effects across the whole population and look at how outcomes would change under possible interventions. In contrast, we focus on explaining what caused a particular observed outcome. By conditioning on the observed treatments and outcome, we define a set of posterior causal estimands that allow for retrospective and individual-level explanations. This can be especially useful in applications where personalized decisions or responsibility need to be considered, such as in medicine or legal settings. Our method is complementary to these prior approaches, offering a distinct and useful perspective on causal inference with continuous outcomes.

In this paper, we focused on the posterior total, direct, and indirect effects of multiple causes. An interesting future direction is to look at how specific causal paths affect the outcome. For example, one may want to know how much of the effect is transmitted through a certain pathway, such as through a known mediator. This idea, known as path-specific effect, has been discussed in earlier work by Pearl (2001). Extending our method to handle such effects could help provide more detailed explanations, especially in settings where causal pathways are well established.

Beyond the path-specific effects, there are also several potential directions for future research. Firstly, in addition to continuous outcomes, the attribution analysis of continuous causes is common in practice and is an issue of great interest. Second, the problem of retrospective analysis when multiple causes and multiple continuous outcomes are involved in many medical diagnoses is also worth exploring (Li et al., 2023). Finally, it is also interesting to discuss the bounds of the proposed estimands or to consider sensitivity analysis when the monotonicity assumption does not hold (Tian & Pearl, 2000; Dawid et al., 2024). These issues are beyond the scope of this paper and are left for future research.

## Acknowledgements

We appreciate the comments from anonymous reviewers, which greatly helped to improve the paper. The authors are supported by the National Natural Science Foundation of China (12401378, 62306019), the Beijing Key Laboratory of Applied Statistics and Digital Regulation, the BTBU Digital Business Platform Project by BMEC, and the Research Foundation for Youth Scholars of Beijing Technology and Business University (RFYS2025).

## Impact Statement

This paper proposes a method for causal attribution analysis in the context of continuous outcome variables, which has the potential to improve decision-making in fields such as medicine and policy analysis. By offering a more accurate framework for identifying and estimating posterior intervention effects, our work may lead to more reliable conclusions in various practical scenarios. Ethical considerations include ensuring the robustness and fairness of causal conclusions, especially in domains affecting public health and social well-being. We believe that the methods developed here could contribute to a deeper understanding of complex causal relationships in real-world data.

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

# Supplementary Material

The Supplementary Material contains proofs of all theoretical results, identifiability results under a causal network, simulation details of the proposed procedure, and a real data analysis.

## S1. Comparison of posterior causal estimands

This section provides the definitions of posterior causal estimands under both continuous and binary outcomes. Let $Y^*$ denote a binary outcome, and let $Y^*_{X=x}$ represent the potential outcome under intervention $X = x$. Following the framework of Lu et al. (2023), Table S3 clearly illustrates the notational correspondence between our proposed estimands and their binary counterparts. While the definitions of $\mathrm{PostNDE}^*$ and $\mathrm{PostNIE}^*$ in Table S3 are conceptually consistent with their continuous counterparts, their identifiability in the binary outcome setting has not been formally established in the existing literature.

Table S3: Definitions of posterior causal estimands under continuous and binary outcomes.

| Estimand | Outcome Type | Definition |
|---|---|---|
| PostTCE | Continuous | $\mathrm{PostTCE}(X_k \Rightarrow Y \mid x, \mathcal{E}) = \mathbb{E}(Y_{X_k=1} - Y_{X_k=0} \mid x, \mathcal{E})$ |
| | Binary | $\mathrm{PostTCE}^*(X_k \Rightarrow \mathcal{E} \mid x, Y^* = 1) = \mathbb{E}(Y^*_{X_k=1} - Y^*_{X_k=0} \mid x, Y^* = 1)$ |
| PostNDE | Continuous | $\mathrm{PostNDE}(X_k \Rightarrow Y \mid x, \mathcal{E}) = \mathbb{E}\{Y_{X_k=1,D_k(a_k,0)} - Y_{X_k=0} \mid x, \mathcal{E}\}$ |
| | Binary | $\mathrm{PostNDE}^*(X_k \Rightarrow \mathcal{E} \mid x, Y^* = 1) = \mathbb{E}\{Y^*_{X_k=1,D_k(a_k,0)} - Y^*_{X_k=0} \mid x, Y^* = 1\}$ |
| PostNIE | Continuous | $\mathrm{PostNIE}(X_k \Rightarrow Y \mid x, \mathcal{E}) = \mathbb{E}\{Y_{X_k=1} - Y_{X_k=1,D_k(a_k,0)} \mid x, \mathcal{E}\}$ |
| | Binary | $\mathrm{PostNIE}^*(X_k \Rightarrow \mathcal{E} \mid x, Y^* = 1) = \mathbb{E}\{Y^*_{X_k=1} - Y^*_{X_k=1,D_k(a_k,0)} \mid x, Y^* = 1\}$ |
| PostICE | Continuous | $\mathrm{PostICE}(Y_{x'} \mid x, \mathcal{E}) = \mathbb{E}(Y_{x'} - Y \mid x, \mathcal{E})$ |
| | Binary | $\mathrm{PostICE}^*(Y^*_{x'} \mid x, Y^* = 1) = \mathbb{E}(Y^*_{x'} - Y^* \mid x, Y^* = 1)$ |
| ITE | Continuous | $\mathrm{ITE}(x', x^*) = Y_{x'} - Y_{x^*}$ |
| | Binary | $\mathrm{ITE}(x', x^*) = Y^*_{x'} - Y^*_{x^*}$ |

From a practical perspective, the choice of which estimand to use depends on the specific analytical goal. For example, given the observed evidence, PostTCE is appropriate when the interest lies in evaluating the overall effect of a single cause, while PostNDE and PostNIE are useful when disentangling direct and indirect pathways is of primary interest. PostICE may be preferred in settings involving multiple simultaneous interventions, where understanding the joint contribution of correlated causes is crucial. Finally, ITE provides individual-level contrast and is most relevant in personalized decision-making contexts, such as precision medicine or targeted policy analysis.

## S2. The proof of Lemma 3.4

*Proof.* For any $x \neq x'$, we recall that the definitions of two marginal distributions $F_x(y)$ and $F_{x'}(y)$ are $F_x(y) = \mathrm{pr}(Y \leq y \mid X = x)$ and $F_{x'}(y) = \mathrm{pr}(Y \leq y \mid X = x')$. Let $F_{\epsilon_{p+1}}(t) = \mathrm{pr}(\epsilon_{p+1} \leq t)$ denote the marginal distribution of the error term $\epsilon_{p+1}$. Let $\mathcal{S}_{\epsilon_{p+1}}$ be the support of the error term $\epsilon_{p+1}$. Since the function $m_{p+1}(\cdot, \epsilon_{p+1})$ is strictly monotonic in $\epsilon_{p+1}$ under Assumption 3.3, for any $\tau \in \mathcal{S}^{\circ}_{\epsilon_{p+1}}$ (the interior of $\mathcal{S}_{\epsilon_{p+1}}$), we have,

$$F_{\epsilon_{p+1}}(\tau) = F_{x'}\{m_{p+1}(x', \tau)\} = F_x\{m_{p+1}(x, \tau)\}.$$

Hence, we have $m_{p+1}(x', \tau) \in \mathcal{S}^{\circ}_{Y_{x'}}$. By the fact that the marginal distribution $F_{x'}(\cdot)$ is continuous and strictly increasing at $m_{p+1}(x', \tau)$, we have,

$$m_{p+1}(x', \tau) = F_{x'}^{-1}\left[F_x\{m_{p+1}(x, \tau)\}\right].$$

Let $y = m_{p+1}(x, \tau) \in \mathcal{S}^{\circ}_{Y_x}$, and we have $\tau = m_{p+1}^{-1}(x, y)$. Then the above equation becomes

$$m_{p+1}(x', \tau) = F_{x'}^{-1}\{F_x(y)\},$$

which shows that $\phi_{x \to x'}(y)$ is identified on $\mathcal{S}_{Y_x}^{\circ}$ (the interior of $\mathcal{S}_{Y_x}$) by

$$\phi_{x \to x'}(y) = m_{p+1}(x', \tau) = F_{x'}^{-1}\{F_x(y)\}.$$

The function $\phi_{x \to x'}(y)$ is identified on $\mathcal{S}_{Y_x}$ by its continuous extension. The remaining results follow immediately. $\square$

## S3. The proof of Lemma 3.5

*Proof.* Without loss of generality, let $x_{k+1:k} = x_{k+1:k}^* = \emptyset$. Let $X_{j:l} = (X_j, \ldots, X_l)$ and $x_{j:l} = (x_j, \ldots, x_l)$. We first simplify the expression of the conditional probability as follows:

$$
\begin{aligned}
&\mathrm{pr}\{D_k(a_k, x_k^*) = d_k^* \mid x\} \\
=&\mathrm{pr}\{D_k(a_k, x_k^*) = d_k^* \mid D_k(a_k, x_k) = d_k\} \\
=&\frac{\mathrm{pr}\{D_k(a_k, x_k^*) = d_k^*, D_k(a_k, x_k) = d_k\}}{\mathrm{pr}\{D_k(a_k, x_k) = d_k\}} \\
=&\frac{\mathrm{pr}\{X_{k+1}(a_k, x_k^*) = x_{k+1}^*, \ldots, X_p(a_{p-1}, x_{k:p-1}^*) = x_p^*, X_{k+1}(a_k, x_k) = x_{k+1}, \ldots, X_p(a_{p-1}, x_{k:p-1}) = x_p\}}{\mathrm{pr}\{X_{k+1}(a_k, x_k) = x_{k+1}, \ldots, X_p(a_{p-1}, x_{k:p-1}) = x_p\}} \\
=&\frac{\mathrm{pr}\{X_{k+1}(a_k, x_k^*) = x_{k+1}^*, X_{k+1}(a_k, x_k) = x_{k+1}\} \times \ldots \times \mathrm{pr}\{X_p(a_{p-1}, x_{k:p-1}^*) = x_p^*, X_p(a_{p-1}, x_{k:p-1}) = x_p\}}{\mathrm{pr}\{X_{k+1}(a_k, x_k) = x_{k+1}\} \times \ldots \times \mathrm{pr}\{X_p(a_{p-1}, x_{k:p-1}) = x_p\}} \\
=&\textstyle\prod_{s=k+1}^{p} \mathrm{pr}\{X_s(a_k, x_{k:s-1}^*) = x_s^* \mid X_s(a_k, x_{k:s-1}) = x_s\}.
\end{aligned}
$$

where the second-to-last equation holds due to Assumption 3.1.

We first consider the case (i), namely $(x_k^*, x_k) = (0, 1)$ and $d_k^* = (x_{k+1}^*, \ldots, x_p^*) \preceq d_k = (x_{k+1}, \ldots, x_p)$. Under the monotonicity assumption 3.2, we have,

$$
\begin{aligned}
\mathrm{pr}\{X_s(a_k, 0, x_{k+1:s-1}^*) = 1 \mid X_s(a_k, 1, x_{k+1:s-1}) = 1\} &= \frac{\mathrm{pr}\{X_s(a_k, 0, x_{k+1:s-1}^*) = 1, X_s(a_k, 1, x_{k+1:s-1}) = 1\}}{\mathrm{pr}\{X_s(a_k, 1, x_{k+1:s-1}) = 1\}} \\
&= \frac{\mathrm{pr}\{X_s(a_k, 0, x_{k+1:s-1}^*) = 1\}}{\mathrm{pr}\{X_s(a_k, 1, x_{k+1:s-1}) = 1\}} \\
&= \frac{\mathrm{pr}(X_s = 1 \mid a_k, 0, x_{k+1:s-1}^*)}{\mathrm{pr}(X_s = 1 \mid a_k, 1, x_{k+1:s-1})}. \\
\mathrm{pr}\{X_s(a_k, 0, x_{k+1:s-1}^*) = 0 \mid X_s(a_k, 1, x_{k+1:s-1}) = 1\} &= 1 - \mathrm{pr}\{X_s(a_k, 0, x_{k+1:s-1}^*) = 1 \mid X_s(a_k, 1, x_{k+1:s-1}) = 1\}. \\
\mathrm{pr}\{X_s(a_k, 0, x_{k+1:s-1}^*) = 0 \mid X_s(a_k, 1, x_{k+1:s-1}) = 0\} &= \frac{\mathrm{pr}\{X_s(a_k, 0, x_{k+1:s-1}^*) = 0, X_s(a_k, 1, x_{k+1:s-1}) = 0\}}{\mathrm{pr}\{X_s(a_k, 1, x_{k+1:s-1}) = 0\}} \\
&= \frac{\mathrm{pr}\{X_s(a_k, 1, x_{k+1:s-1}) = 0\}}{\mathrm{pr}\{X_s(a_k, 1, x_{k+1:s-1}) = 0\}} \\
&= 1. \\
\mathrm{pr}\{X_s(a_k, 0, x_{k+1:s-1}^*) = 1 \mid X_s(a_k, 1, x_{k+1:s-1}) = 0\} &\\
&= 1 - \mathrm{pr}\{X_s(a_k, 0, x_{k+1:s-1}^*) = 0 \mid X_s(a_k, 1, x_{k+1:s-1}) = 0\} \\
&= 0.
\end{aligned}
$$

Therefore, we have,

$$\mathrm{pr}\{X_s(a_k, 0, x_{k+1:s-1}^*) = x_s^* \mid X_s(a_k, 1, x_{k+1:s-1}) = x_s\} = (1 - x_s^*) + (2x_s^* - 1)x_s R_{0s},$$

where

$$R_{0s} = \frac{\mathrm{pr}(X_s = 1 \mid a_k, 0, x_{k+1}^*, \ldots, x_{s-1}^*)}{\mathrm{pr}(X_s = 1 \mid a_k, 1, x_{k+1}, \ldots, x_{s-1})}.$$

We next consider the case (ii), namely $(x_k^*, x_k) = (1, 0)$ and $d_k = (x_{k+1}, \ldots, x_p) \preceq d_k^* = (x_{k+1}^*, \ldots, x_p^*)$. Under the

monotonicity assumption 3.2, we have,

$$\text{pr}\{X_s(a_k, 1, x^*_{k+1:s-1}) = 0 \mid X_s(a_k, 0, x_{k+1:s-1}) = 0\} = \frac{\text{pr}\{X_s(a_k, 1, x^*_{k+1:s-1}) = 0, X_s(a_k, 0, x_{k+1:s-1}) = 0\}}{\text{pr}\{X_s(a_k, 0, x_{k+1:s-1}) = 0\}}$$

$$= \frac{\text{pr}\{X_s(a_k, 1, x^*_{k+1:s-1}) = 0\}}{\text{pr}\{X_s(a_k, 0, x_{k+1:s-1}) = 0\}}$$

$$= \frac{\text{pr}(X_s = 0 \mid a_k, 1, x^*_{k+1:s-1})}{\text{pr}(X_s = 0 \mid a_k, 0, x_{k+1:s-1})}.$$

$$\text{pr}\{X_s(a_k, 1, x^*_{k+1:s-1}) = 1 \mid X_s(a_k, 0, x_{k+1:s-1}) = 0\} = 1 - \text{pr}\{X_s(a_k, 1, x^*_{k+1:s-1}) = 0 \mid X_s(a_k, 0, x_{k+1:s-1}) = 0\}.$$

$$\text{pr}\{X_s(a_k, 1, x^*_{k+1:s-1}) = 1 \mid X_s(a_k, 0, x_{k+1:s-1}) = 1\} = \frac{\text{pr}\{X_s(a_k, 1, x^*_{k+1:s-1}) = 1, X_s(a_k, 0, x_{k+1:s-1}) = 1\}}{\text{pr}\{X_s(a_k, 0, x_{k+1:s-1}) = 1\}}$$

$$= \frac{\text{pr}\{X_s(a_k, 0, x_{k+1:s-1}) = 1\}}{\text{pr}\{X_s(a_k, 0, x_{k+1:s-1}) = 1\}}$$

$$= 1.$$

$$\text{pr}\{X_s(a_k, 1, x^*_{k+1:s-1}) = 0 \mid X_s(a_k, 0, x_{k+1:s-1}) = 1\}$$
$$= 1 - \text{pr}\{X_s(a_k, 1, x^*_{k+1:s-1}) = 1 \mid X_s(a_k, 0, x_{k+1:s-1}) = 1\}$$
$$= 0.$$

Therefore, we have,

$$\text{pr}\{X_s(a_k, 1, x^*_{k+1:s-1}) = x^*_s \mid X_s(a_k, 0, x_{k+1:s-1}) = x_s\} = x^*_s + (1 - 2x^*_s)(1 - x_s)R_{1s},$$

where

$$R_{1s} = \frac{\text{pr}(X_s = 0 \mid a_k, 1, x^*_{k+1}, \ldots, x^*_{s-1})}{\text{pr}(X_s = 0 \mid a_k, 0, x_{k+1}, \ldots, x_{s-1})}.$$

$\square$

## S4. The proof of Theorem 3.6

*Proof.* When $x_k = 1$, we first consider $E\{Y_{a_k, x^*_k, D_k(a_k, 1)}\}$:

$$E\{Y_{a_k, x^*_k, D_k(a_k, 1)} \mid a_k, 1, d_k, Y \in \mathcal{E}\} = E\{Y_{a_k, x^*_k, D_k(a_k, 1)} \mid a_k, 1, D_k(a_k, 1) = d_k, Y \in \mathcal{E}\}$$
$$= E\{Y_{a_k, x^*_k, d_k} \mid a_k, 1, D_k(a_k, 1) = d_k, Y \in \mathcal{E}\}$$
$$= E(Y_{a_k, x^*_k, d_k} \mid a_k, 1, d_k, Y \in \mathcal{E}).$$

We next consider $E\{Y_{a_k, x^*_k, D_k(a_k, 0)}\}$:

$$E\{Y_{a_k, x^*_k, D_k(a_k, 0)} \mid a_k, 1, d_k, Y \in \mathcal{E}\}$$
$$= \sum_{d^*_k \preceq d_k} E\{Y_{a_k, x^*_k, d^*_k} \mid a_k, 1, d_k, D_k(a_k, 0) = d^*_k, Y \in \mathcal{E}\}\text{pr}\{D_k(a_k, 0) = d^*_k \mid a_k, 1, d_k\}$$
$$= \sum_{d^*_k \preceq d_k} E(Y_{a_k, x^*_k, d^*_k} \mid a_k, 1, d_k, Y \in \mathcal{E})\text{pr}\{D_k(a_k, 0) = d^*_k \mid a_k, 1, d_k\},$$

where the second equality holds due to Assumption 3.1 (ii) and Assumption 3.2. The proofs for $x_k = 0$ follow a similar logic; for simplicity, we omit them. $\square$

## S5. The proof of Corollary 3.7

*Proof.* Note that because $m_{p+1}(X, \cdot)$ is strictly monotone in $\epsilon_{p+1}$, for any $\tau \in \mathcal{S}^\circ_{\epsilon_{p+1}}$ (the interior of $\mathcal{S}_\epsilon$) and for any $\text{pa}^*_y \neq \text{pa}'_y$, we have:

$$F_{\epsilon_{p+1}}(\tau) = F_{\text{pa}^*_y}\{m_{p+1}(\text{pa}^*_y, \tau)\} = F_{\text{pa}'_y}\{m_{p+1}(\text{pa}'_y, \tau)\}.$$

The remaining proof follows a similar logic as Section S2; we omit it for simplicity. $\square$

## S6. The proof of Corollary 3.8

*Proof.* Given the Bayesian network, the distribution of each node depends only on its parent nodes. The proof is straightforward, and we omit it for brevity. □

## S7. The proof of Lemma 4.2

*Proof.* First, we differentiate $\rho_{x\to x'}(y_{x'}; y_x)$ with respect to $y_x$. Noting that

$$\frac{\partial E(|Y - y| \mid X = x)t}{\partial t} = -E\{\text{sign}(Y - y) \mid X = x\} = 1 - 2\text{pr}(Y < y \mid X = x) = 1 - 2F_x(y),$$

$$\frac{\partial E(|Y - t| \mid X = x')}{\partial t} = -E\{\text{sign}(Y - t) \mid X = x'\} = 1 - 2\text{pr}(Y < t \mid X = x') = 1 - 2F_{x'}(t),$$

where $\text{sign}(u) \equiv 2 \times \mathbb{I}(u > 0) - 1$. It follows that

$$\frac{\partial \rho_{x\to x'}(t; y)}{\partial t} = 2\{F_{x'}(t) - F_x(y)\} = 0.$$

Fix $y_x \in \mathbb{R}$. Note that the marginal distribution $F_{x'}(\cdot)$ is weakly increasing on $\mathbb{R}$ and strictly increasing on $\mathcal{S}_{Y_{x'}}^\circ$. Therefore, $\rho_{x\to x'}(\cdot; y_x)$ is weakly and strictly convex on $\mathbb{R}$ and $\mathcal{S}_{Y_x}^\circ$. Furthermore, under Assumption 3.3, we know that $\phi_{x\to x'}(y_x) = F_{x'}^{-1}\{F_x(y_x)\}$. For $y_{x'} \in \mathcal{S}_{Y_{x'}}^\circ$, we have $F_x(y_x) = F_{x'}(y_{x'})$ if and only if $y_{x'} = \phi_{x\to x'}(y_{x'})$. Thus, $y_{x'} = \phi_{x\to x'}(y_x)$ uniquely solves the first-order condition $\frac{\partial}{\partial t}\rho_{x\to x'}(t; y_x) = 0$. □

## S8. Additional analysis for hypertension example in Section 5

In this section, we present additional analysis conclusions under different evidence. Given the observed evidence $\{X = (1, 1, 1, 0, 1), Y > 140\}$, Table S4 presents the results for the posterior causal estimands with respect to each possible risk factor $X_k$. The first two rows of results are directly obtained from Lu et al. (2023) for comparison purposes. In the second row, we observe that $\text{postTCE}^*(X_1 \Rightarrow \mathcal{E} \mid \cdot) = 0.2$, and $\text{postTCE}^*(X_k \Rightarrow \mathcal{E} \mid \cdot) = 0$ for $k = 2, \ldots, 5$. Our results regarding postTCE are presented in the fifth row, showing a significant increase in blood pressure by 10.483 units when suffering from heart disease, i.e., $\text{postTCE}(X_4 \Rightarrow Y \mid \cdot) = 10.483$. Our results differ from Lu et al. (2023) with respect to $X_4$. This is because Lu et al. (2023) additionally requires the monotonicity assumption of the outcome variable for identifiability in the binary outcome case. This assumption ensures that given the evidence $X_4 = 0$ (indicating no heart disease), $X_1$ is the unique risk factor. However, the perfect rank assumption 3.3 introduced for the continuous variable does not guarantee this. It can also be observed that the probability of necessity is not zero for $X_2$, $X_3$, and $X_5$; $\text{postTCE}^*(X_k \Rightarrow \mathcal{E} \mid \cdot)$, $\text{postTCE}(X_k \Rightarrow Y \mid \cdot)$, and $\text{postNDE}(X_k \Rightarrow Y \mid \cdot)$ are zero when $X_4 = 0$ for $k = 1, 2, 5$. Additionally, it can be observed from the fourth row that $\text{postNIE}(X_k \Rightarrow Y \mid \cdot) = 0$ for all $k = 1, \ldots, 5$, suggesting that all causes have no indirect effect on BP. From the causal network in Figure 1, since the evidence indicates no heart disease, i.e., $X_4 = 0$, it can be intuitively understood that $X_4$ blocks other nodes from transmitting effects to the outcome along the paths.

Table S4: Results of marginal probabilities of necessity and posterior causal estimands based on the evidence $\{X = (1, 1, 1, 0, 1), Y > 140\}$.

|  | $X_1$ | $X_2$ | $X_3$ | $X_4$ | $X_5$ |
|---|---|---|---|---|---|
| $\text{PN}^*(X_k \Rightarrow \mathcal{E})$ | 0.347 | 0.230 | 0.133 | # | 0.563 |
| $\text{postTCE}^*(X_k \Rightarrow \mathcal{E} \mid x, Y > 140)$ | 0.200 | 0 | 0 | 0 | 0 |
| $\text{postNDE}(X_k \Rightarrow Y \mid x, Y > 140)$ | 2.000 | 0 | 0 | 10.483 | 0 |
| $\text{postNIE}(X_k \Rightarrow Y \mid x, Y > 140)$ | 0 | 0 | 0 | 0 | 0 |
| $\text{postTCE}(X_k \Rightarrow Y \mid x, Y > 140)$ | 2.000 | 0 | 0 | 10.483 | 0 |

\# Corresponding quantity is undefined.

For the observed evidence $\{X = (1, 0, 1, 1, 1), Y > 140\}$, where $X_2 = 0$ indicates a healthy diet, the values of postTCE and postNDE are shown in Table S5. Comparing it with Table 2, we see that

$$\text{postTCE}\{X_1 \Rightarrow Y \mid X = (1, 0, 1, 1, 1), Y > 140\} = 12.418 > \text{postTCE}\{X_1 \Rightarrow Y \mid X = (1, 1, 1, 1, 1), Y > 140\} = 10.628,$$

Table S5: Results of marginal probabilities of necessity and posterior causal estimands based on the evidence $\{X = (1, 0, 1, 1, 1), Y > 140\}$.

|  | $X_1$ | $X_2$ | $X_3$ | $X_4$ | $X_5$ |
|---|---|---|---|---|---|
| postTCE$^*$ $(X_k \Rightarrow \mathcal{E} \mid x, Y > 140)$ | 0.449 | 0 | 0 | 0.722 | 0 |
| postNDE $(X_k \Rightarrow Y \mid x, Y > 140)$ | 3.424 | 0 | 0 | 16.856 | 0 |
| postNIE $(X_k \Rightarrow Y \mid x, Y > 140)$ | 8.994 | 0 | 0 | 0 | 0 |
| postTCE $(X_k \Rightarrow Y \mid x, Y > 140)$ | 12.418 | 0 | 0 | 16.856 | 0 |

and

$$\text{postTCE}\{X_2 \Rightarrow Y \mid X = (1, 0, 1, 1, 1), Y > 140\} = 0 < \text{postTCE}\{X_2 \Rightarrow Y \mid X = (1, 1, 1, 1, 1), Y > 140\} = 4.561.$$

This is because changing $X_2 = 1$ to $X_2 = 0$ in the evidence increases the possibility that $X_1$ is the cause and decreases the possibility that $X_2$ is the cause. Similar conclusions are also found in Lu et al. (2023) in the binary case.

## S9. Simulation studies for hypertension example in Section 5

### S9.1. Data generating details

In this section, we provide additional simulation studies related to Section 5. To generate simulation data that satisfies Figure 1 in Lu et al. (2023), we consider the following data generation process.

(a) $X_1$ is Bernoulli with $\text{pr}(X_1 = 1) = 0.3$.

(b) $X_2$ is Bernoulli with $\text{pr}(X_2 = 1) = 0.75$.

(c) Let $G_3 = \{X_3(X_2 = 1), X_3(X_2 = 0)\}$, then $G_3$ can take values on 00, 01, and 11 under monotonicity assumption 3.2,

$$\text{pr}(G_3 = 00) = 0.15, \ \text{pr}(G_3 = 01) = 0.65, \ \text{pr}(G_3 = 11) = 0.2.$$

Here, $X_3$ is generated as follows:

$$X_3 = 0 \text{ if } G_3 = 00,$$
$$X_3 = 1 \text{ if } G_3 = 11,$$
$$X_3 = 0 \text{ if } G_3 = 01, X_2 = 0,$$
$$X_3 = 0 \text{ if } G_3 = 01, X_2 = 1.$$

(iv) Let $G_4 = \{X_3(X_1 = 0, X_2 = 0), X_3(X_1 = 0, X_2 = 1), X_3(X_1 = 1, X_2 = 0), X_3(X_1 = 1, X_2 = 1)\}$, then $G_4$ can take values on 0000, 0001, 0011, 0101, 0111, and 1111 under monotonicity assumption 3.2,

$$\text{pr}(G_4 = 0000) = 0.25, \ \text{pr}(G_4 = 0001) = 0.10, \ \text{pr}(G_4 = 0011) = 0.20,$$
$$\text{pr}(G_4 = 0101) = 0.10, \ \text{pr}(G_4 = 0111) = 0.10, \ \text{pr}(G_4 = 1111) = 0.25.$$

Here, $X_4$ is generated as follows:

$$
\begin{aligned}
&X_4 = 0 \text{ if } G_4 = 0000, \\
&X_4 = 1 \text{ if } G_4 = 0001, X_1 = 1, X_2 = 1, \\
&X_4 = 0 \text{ if } G_4 = 0001, X_1 = 0, X_2 = 1, \\
&X_4 = 0 \text{ if } G_4 = 0001, X_1 = 1, X_2 = 0, \\
&X_4 = 0 \text{ if } G_4 = 0001, X_1 = 0, X_2 = 0, \\
&X_4 = 1 \text{ if } G_4 = 0011, X_1 = 1, \\
&X_4 = 0 \text{ if } G_4 = 0011, X_1 = 0, \\
&X_4 = 1 \text{ if } G_4 = 0101, X_2 = 1, \\
&X_4 = 0 \text{ if } G_4 = 0101, X_2 = 0, \\
&X_4 = 1 \text{ if } G_4 = 0111, X_1 = 1, X_2 = 1, \\
&X_4 = 1 \text{ if } G_4 = 0111, X_1 = 0, X_2 = 1, \\
&X_4 = 1 \text{ if } G_4 = 0111, X_1 = 1, X_2 = 0, \\
&X_4 = 0 \text{ if } G_4 = 0111, X_1 = 0, X_2 = 0, \\
&X_4 = 1 \text{ if } G_4 = 1111.
\end{aligned}
$$

(d) Let $G_5 = \{X_3(X_3 = 0, X_4 = 0), X_3(X_3 = 0, X_4 = 1), X_3(X_3 = 1, X_4 = 0), X_3(X_3 = 1, X_4 = 1)\}$, then $G_5$ can take values on 0000, 0001, 0011, 0101, 0111, and 1111 under monotonicity assumption 3.2,

$$
\begin{aligned}
&\text{pr}(G_5 = 0000) = 0.10, \ \text{pr}(G_5 = 0001) = 0.05, \ \text{pr}(G_5 = 0011) = 0.45, \\
&\text{pr}(G_5 = 0101) = 0.05, \ \text{pr}(G_5 = 0111) = 0.25, \ \text{pr}(G_5 = 1111) = 0.10.
\end{aligned}
$$

Here, $X_5$ is generated as follows:

$$
\begin{aligned}
&X_5 = 0 \text{ if } G_5 = 0000, \\
&X_5 = 1 \text{ if } G_5 = 0001, X_3 = 1, X_4 = 1, \\
&X_5 = 0 \text{ if } G_5 = 0001, X_3 = 0, X_4 = 1, \\
&X_5 = 0 \text{ if } G_5 = 0001, X_3 = 1, X_4 = 0, \\
&X_5 = 0 \text{ if } G_5 = 0001, X_3 = 0, X_4 = 0, \\
&X_5 = 1 \text{ if } G_5 = 0011, X_3 = 1, \\
&X_5 = 0 \text{ if } G_5 = 0011, X_3 = 0, \\
&X_5 = 1 \text{ if } G_5 = 0101, X_4 = 1, \\
&X_5 = 0 \text{ if } G_5 = 0101, X_4 = 0, \\
&X_5 = 1 \text{ if } G_5 = 0111, X_3 = 1, X_4 = 1, \\
&X_5 = 1 \text{ if } G_5 = 0111, X_3 = 0, X_4 = 1, \\
&X_5 = 1 \text{ if } G_5 = 0111, X_3 = 1, X_4 = 0, \\
&X_5 = 0 \text{ if } G_5 = 0111, X_3 = 0, X_4 = 0, \\
&X_5 = 1 \text{ if } G_5 = 1111.
\end{aligned}
$$

The above data generation process will result in an observation distribution exactly identical to Lu et al. (2023) and will be utilized to compute the true values of posterior causal estimands considered in this paper.

**S9.2. The simulation results for posterior causal estimands**

In this section, we conducted simulation studies to assess the performance of the proposed procedure in finite samples. We generated numerical examples corresponding to Figure 1 of the main text using the method described in Section S9.1. Using data generated from 2000000 samples, we obtained the estimated results presented in Tables 2, S4, and S5 of the main text,

as well as the true values provided in Tables S6 and S7 of the Supplementary Material. Additionally, we considered sample sizes of $n = 1000, 2000$, and $10000$, and averaged the results over 500 repetitions. These simulation results demonstrate the stability of the proposed identification expressions. Specifically, we observed that the estimated values are close to the true values, and the standard errors are relatively small. As the sample size increases, both the bias and standard errors decrease.

(i) Table S6 presents the true values and estimated results of the posterior intervention causal effect based on different evidence.

(ii) Table S7 presents the true values of the posterior natural direct and indirect causal effect based on different evidence.

(iii) Tables S8-S10 present the estimated results of the posterior direct effect based on different evidence.

(iv) Tables S11-S13 present the estimated results of the posterior indirect effect based on different evidence.

Table S6: Posterior intervention causal effect based on different pieces of evidence.

|  | PostICE$(Y'_x \mid x, Y > 140)$ | $(x_1, x_4) = (0,0)$ | $(x_1, x_4) = (0,1)$ | $(x_1, x_4) = (1,0)$ | $(x_1, x_4) = (1,1)$ |
|---|---|---|---|---|---|
| True values | $(x'_1, x'_4) = (0,0)$ | 0 | -13.26 | -2 | -19 |
|  | $(x'_1, x'_4) = (0,1)$ | 3.50 | 0 | 2.25 | -5 |
|  | $(x'_1, x'_4) = (1,0)$ | 2 | -11.26 | 0 | -17 |
|  | $(x'_1, x'_4) = (1,1)$ | 12 | 5.25 | 10.5 | 0 |
| $n = 1000$ | $(x'_1, x'_4) = (0,0)$ | 0.000 (0.000) | -13.252 (0.722) | -1.757 (1.210) | -18.955 (0.769) |
|  | $(x'_1, x'_4) = (0,1)$ | 3.477 (0.317) | 0.000 (0.000) | 2.391 (0.699) | -4.998 (0.432) |
|  | $(x'_1, x'_4) = (1,0)$ | 1.561 (1.129) | -11.532 (1.496) | 0.000 (0.000) | -17.224 (1.480) |
|  | $(x'_1, x'_4) = (1,1)$ | 11.967 (0.435) | 5.227 (0.518) | 10.676 (0.835) | 0.000 (0.000) |
| $n = 2000$ | $(x'_1, x'_4) = (0,0)$ | 0.000 (0.000) | -13.224 (0.554) | -1.921 (0.928) | -18.962 (0.602) |
|  | $(x'_1, x'_4) = (0,1)$ | 3.463 (0.216) | 0.000 (0.000) | 2.273 (0.495) | -4.997 (0.300) |
|  | $(x'_1, x'_4) = (1,0)$ | 1.808 (0.862) | -11.322 (1.031) | 0.000 (0.000) | -17.057 (1.073) |
|  | $(x'_1, x'_4) = (1,1)$ | 11.944 (0.296) | 5.240 (0.361) | 10.518 (0.631) | 0.000 (0.000) |
| $n = 10000$ | $(x'_1, x'_4) = (0,0)$ | 0.000 (0.0000) | -13.261 (0.2636) | -1.950 (0.3588) | -19.036 (0.2550) |
|  | $(x'_1, x'_4) = (0,1)$ | 3.503 (0.0992) | 0.000 (0.0000) | 2.291 (0.1965) | -5.005 (0.1383) |
|  | $(x'_1, x'_4) = (1,0)$ | 1.926 (0.3298) | -11.307 (0.4901) | 0.000 (0.0000) | -17.076 (0.4652) |
|  | $(x'_1, x'_4) = (1,1)$ | 11.995 (0.1558) | 5.261 (0.1576) | 10.543 (0.2530) | 0.000 (0.0000) |

Table S7: True values of posterior natural direct effect and posterior natural indirect effect.

| Case | Evidence $(x,\mathcal{E})$ | Posterior natural direct effect | | | | | Posterior natural indirect effect | | | | |
|---|---|---|---|---|---|---|---|---|---|---|---|
| | | $X_1$ | $X_2$ | $X_3$ | $X_4$ | $X_5$ | $X_1$ | $X_2$ | $X_3$ | $X_4$ | $X_5$ |
| 1 | $(0,0,0,0,0,\mathcal{E})$ | 2.000 | 0.000 | 0.000 | 3.482 | 0.000 | 3.885 | 0.918 | 0.000 | 0.000 | 0.000 |
| 2 | $(0,0,0,0,1,\mathcal{E})$ | 2.000 | 0.000 | 0.000 | 3.331 | 0.000 | 3.694 | 0.796 | 0.000 | 0.000 | 0.000 |
| 3 | $(0,0,0,1,0,\mathcal{E})$ | 5.197 | 0.000 | 0.000 | 13.408 | 0.000 | 0.000 | 0.000 | 0.000 | 0.000 | 0.000 |
| 4 | $(0,0,0,1,1,\mathcal{E})$ | 5.201 | 0.000 | 0.000 | 13.396 | 0.000 | 0.000 | 0.000 | 0.000 | 0.000 | 0.000 |
| 5 | $(0,0,1,0,0,\mathcal{E})$ | 2.000 | 0.000 | 0.000 | 3.539 | 0.000 | 4.460 | 0.992 | 0.000 | 0.000 | 0.000 |
| 6 | $(0,0,1,0,1,\mathcal{E})$ | 2.000 | 0.000 | 0.000 | 3.434 | 0.000 | 3.908 | 0.838 | 0.000 | 0.000 | 0.000 |
| 7 | $(0,0,1,1,0,\mathcal{E})$ | 5.416 | 0.000 | 0.000 | 12.751 | 0.000 | 0.000 | 0.000 | 0.000 | 0.000 | 0.000 |
| 8 | $(0,0,1,1,1,\mathcal{E})$ | 5.171 | 0.000 | 0.000 | 13.486 | 0.000 | 0.000 | 0.000 | 0.000 | 0.000 | 0.000 |
| 9 | $(0,1,0,0,0,\mathcal{E})$ | 2.000 | 0.000 | 0.000 | 3.455 | 0.000 | 5.655 | 0.000 | 0.000 | 0.000 | 0.000 |
| 10 | $(0,1,0,0,1,\mathcal{E})$ | 2.000 | 0.000 | 0.000 | 3.585 | 0.000 | 5.474 | 0.000 | 0.000 | 0.000 | 0.000 |
| 11 | $(0,1,0,1,0,\mathcal{E})$ | 5.126 | 0.000 | 0.000 | 13.621 | 0.000 | 0.000 | 5.926 | 0.000 | 0.000 | 0.000 |
| 12 | $(0,1,0,1,1,\mathcal{E})$ | 5.291 | 0.000 | 0.000 | 13.127 | 0.000 | 0.000 | 5.950 | 0.000 | 0.000 | 0.000 |
| 13 | $(0,1,1,0,0,\mathcal{E})$ | 2.000 | 0.000 | 0.000 | 3.547 | 0.000 | 5.664 | 0.000 | 0.000 | 0.000 | 0.000 |
| 14 | $(0,1,1,0,1,\mathcal{E})$ | 2.000 | 0.000 | 0.000 | 3.483 | 0.000 | 5.407 | 0.000 | 0.000 | 0.000 | 0.000 |
| 15 | $(0,1,1,1,0,\mathcal{E})$ | 5.192 | 0.000 | 0.000 | 13.425 | 0.000 | 0.000 | 5.874 | 0.000 | 0.000 | 0.000 |
| 16 | $(0,1,1,1,1,\mathcal{E})$ | 5.246 | 0.000 | 0.000 | 13.263 | 0.000 | 0.000 | 5.916 | 0.000 | 0.000 | 0.000 |
| 17 | $(1,0,0,0,0,\mathcal{E})$ | 2.000 | 0.000 | 0.000 | 10.490 | 0.000 | 0.000 | 4.602 | 0.000 | 0.000 | 0.000 |
| 18 | $(1,0,0,0,1,\mathcal{E})$ | 2.000 | 0.000 | 0.000 | 10.365 | 0.000 | 0.000 | 4.864 | 0.000 | 0.000 | 0.000 |
| 19 | $(1,0,0,1,0,\mathcal{E})$ | 3.342 | 0.000 | 0.000 | 16.955 | 0.000 | 9.184 | 0.000 | 0.000 | 0.000 | 0.000 |
| 20 | $(1,0,0,1,1,\mathcal{E})$ | 3.304 | 0.000 | 0.000 | 17.188 | 0.000 | 9.477 | 0.000 | 0.000 | 0.000 | 0.000 |
| 21 | $(1,0,1,0,0,\mathcal{E})$ | 2.000 | 0.000 | 0.000 | 10.569 | 0.000 | 0.000 | 4.793 | 0.000 | 0.000 | 0.000 |
| 22 | $(1,0,1,0,1,\mathcal{E})$ | 2.000 | 0.000 | 0.000 | 10.645 | 0.000 | 0.000 | 3.802 | 0.000 | 0.000 | 0.000 |
| 23 | $(1,0,1,1,0,\mathcal{E})$ | 3.504 | 0.000 | 0.000 | 17.188 | 0.000 | 8.130 | 0.000 | 0.000 | 0.000 | 0.000 |
| 24 | $(1,0,1,1,1,\mathcal{E})$ | 3.424 | 0.000 | 0.000 | 16.856 | 0.000 | 8.994 | 0.000 | 0.000 | 0.000 | 0.000 |
| 25 | $(1,1,0,0,0,\mathcal{E})$ | 2.000 | 0.000 | 0.000 | 10.363 | 0.000 | 0.000 | 0.000 | 0.000 | 0.000 | 0.000 |
| 26 | $(1,1,0,0,1,\mathcal{E})$ | 2.000 | 0.000 | 0.000 | 10.586 | 0.000 | 0.000 | 0.000 | 0.000 | 0.000 | 0.000 |
| 27 | $(1,1,0,1,0,\mathcal{E})$ | 3.867 | 0.000 | 0.000 | 16.809 | 0.000 | 6.740 | 4.306 | 0.000 | 0.000 | 0.000 |
| 28 | $(1,1,0,1,1,\mathcal{E})$ | 3.799 | 0.000 | 0.000 | 17.045 | 0.000 | 6.692 | 4.251 | 0.000 | 0.000 | 0.000 |
| 29 | $(1,1,1,0,0,\mathcal{E})$ | 2.000 | 0.000 | 0.000 | 10.463 | 0.000 | 0.000 | 0.000 | 0.000 | 0.000 | 0.000 |
| 30 | $(1,1,1,0,1,\mathcal{E})$ | 2.000 | 0.000 | 0.000 | 10.483 | 0.000 | 0.000 | 0.000 | 0.000 | 0.000 | 0.000 |
| 31 | $(1,1,1,1,0,\mathcal{E})$ | 3.837 | 0.000 | 0.000 | 16.993 | 0.000 | 6.595 | 4.603 | 0.000 | 0.000 | 0.000 |
| 32 | $(1,1,1,1,1,\mathcal{E})$ | 3.823 | 0.000 | 0.000 | 17.023 | 0.000 | 6.805 | 4.561 | 0.000 | 0.000 | 0.000 |

## S10. Real data analysis: risk factors for abnormal weight

In this section, we apply the proposed method to a real dataset from the developmental toxicology experiments conducted by the National Toxicology Program (NTP) (NTP, 2023). The primary objective of this study is to analyze whether tris(1-chloro-2-propyl) phosphate (TCPP) is a risk factor for abnormal weight loss in B6C3F1/N mice, or if there are other causes that are the potential risks. In this experiment, a total of 120 mice were randomly exposed to six different dose levels of TCPP via dosed feed: 0, 1250, 2500, 5000, 10000, or 20000 ppm, for a duration of 3 months. Each pup's data includes gender (male/female), weekly body weights, organ weights, and whether organs exhibit pathology. In our analysis, let $X_1$ represent the gender of the mice, where $X_1 = 0$ indicates female mice and $X_1 = 1$ indicates male mice. Let $X_2$ denote the dose, where $X_2 = 0$ represents exposure to the low dose group including 0, 1250, and 2500 ppm, and $X_2 = 1$ represents exposure to the high dose group including 5000, 10000, and 20000 ppm. Let $X_3$ denote whether the liver or kidney exhibits pathology, where $X_3 = 0$ indicates no pathology and $X_3 = 1$ indicates pathology. We choose the body weight at the end of three months as the outcome $Y$. In this analysis, we focus on assessing the potential risk factors affecting the body weight of underweight mice, indicated by the event $\mathcal{E} = \mathbb{I}(Y < 27)$.

We first use the R package "bnlearn" to construct a Bayesian network based on the collected data as shown in Figure S2. It is clear that dose $X_2$ indirectly affects body weight $Y$ through the organ disease $X_3$. Since both gender (i.e., $X_1$) and dose (i.e., $X_2$) can be considered as randomized trials, we can estimate the causal effect of $X_1$ on $Y$ using the difference in means estimator, i.e., $E(Y_{X_1=1} - Y_{X_1=0}) \approx 6.09$; which implies that males are heavier. Similarly, we can estimate the causal

Table S8: The simulation results for the postNDE under a sample size of 1000.

| Case | Evidence $(x, \mathcal{E})$ | PostNDE$(X_k \Rightarrow Y \mid x, Y > 140)$ | | | | |
|------|------|------|------|------|------|------|
| | | $X_1$ | $X_2$ | $X_3$ | $X_4$ | $X_5$ |
| 1 | $(0,0,0,0,0,\mathcal{E})$ | 1.561 (1.134) | 0.000 (0.000) | 0.000 (0.000) | 3.489 (0.318) | 0.000 (0.000) |
| 2 | $(0,0,0,0,1,\mathcal{E})$ | 1.561 (1.134) | 0.000 (0.000) | 0.000 (0.000) | 3.489 (0.318) | 0.000 (0.000) |
| 3 | $(0,0,0,1,0,\mathcal{E})$ | 5.226 (0.519) | 0.000 (0.000) | 0.000 (0.000) | 13.255 (0.722) | 0.000 (0.000) |
| 4 | $(0,0,0,1,1,\mathcal{E})$ | 5.226 (0.519) | 0.000 (0.000) | 0.000 (0.000) | 13.255 (0.722) | 0.000 (0.000) |
| 5 | $(0,0,1,0,0,\mathcal{E})$ | 1.561 (1.134) | 0.000 (0.000) | 0.000 (0.000) | 3.489 (0.318) | 0.000 (0.000) |
| 6 | $(0,0,1,0,1,\mathcal{E})$ | 1.561 (1.134) | 0.000 (0.000) | 0.000 (0.000) | 3.489 (0.318) | 0.000 (0.000) |
| 7 | $(0,0,0,1,0,\mathcal{E})$ | 5.226 (0.519) | 0.000 (0.000) | 0.000 (0.000) | 13.255 (0.722) | 0.000 (0.000) |
| 8 | $(0,0,1,1,1,\mathcal{E})$ | 5.226 (0.519) | 0.000 (0.000) | 0.000 (0.000) | 13.255 (0.722) | 0.000 (0.000) |
| 9 | $(0,1,0,0,0,\mathcal{E})$ | 1.561 (1.134) | 0.000 (0.000) | 0.000 (0.000) | 3.489 (0.318) | 0.000 (0.000) |
| 10 | $(0,1,0,0,1,\mathcal{E})$ | 1.561 (1.134) | 0.000 (0.000) | 0.000 (0.000) | 3.489 (0.318) | 0.000 (0.000) |
| 11 | $(0,1,0,1,0,\mathcal{E})$ | 5.226 (0.519) | 0.000 (0.000) | 0.000 (0.000) | 13.255 (0.722) | 0.000 (0.000) |
| 12 | $(0,1,0,1,1,\mathcal{E})$ | 5.226 (0.519) | 0.000 (0.000) | 0.000 (0.000) | 13.255 (0.722) | 0.000 (0.000) |
| 13 | $(0,1,1,0,0,\mathcal{E})$ | 1.561 (1.134) | 0.000 (0.000) | 0.000 (0.000) | 3.489 (0.318) | 0.000 (0.000) |
| 14 | $(0,1,1,0,1,\mathcal{E})$ | 1.561 (1.134) | 0.000 (0.000) | 0.000 (0.000) | 3.489 (0.318) | 0.000 (0.000) |
| 15 | $(0,1,1,1,0,\mathcal{E})$ | 5.226 (0.519) | 0.000 (0.000) | 0.000 (0.000) | 13.255 (0.722) | 0.000 (0.000) |
| 16 | $(0,1,1,1,1,\mathcal{E})$ | 5.226 (0.519) | 0.000 (0.000) | 0.000 (0.000) | 13.255 (0.722) | 0.000 (0.000) |
| 17 | $(1,0,0,0,0,\mathcal{E})$ | 1.755 (1.194) | 0.000 (0.000) | 0.000 (0.000) | 10.641 (0.821) | 0.000 (0.000) |
| 18 | $(1,0,0,0,1,\mathcal{E})$ | 1.756 (1.197) | 0.000 (0.000) | 0.000 (0.000) | 10.641 (0.821) | 0.000 (0.000) |
| 19 | $(1,0,0,1,0,\mathcal{E})$ | 3.230 (0.903) | 0.000 (0.000) | 0.000 (0.000) | 17.220 (1.479) | 0.000 (0.000) |
| 20 | $(1,0,0,1,1,\mathcal{E})$ | 3.230 (0.903) | 0.000 (0.000) | 0.000 (0.000) | 17.220 (1.479) | 0.000 (0.000) |
| 21 | $(1,0,1,0,0,\mathcal{E})$ | 1.755 (1.194) | 0.000 (0.000) | 0.000 (0.000) | 10.641 (0.821) | 0.000 (0.000) |
| 22 | $(1,0,1,0,1,\mathcal{E})$ | 1.753 (1.193) | 0.000 (0.000) | 0.000 (0.000) | 10.641 (0.821) | 0.000 (0.000) |
| 23 | $(1,0,1,1,0,\mathcal{E})$ | 3.371 (1.153) | 0.000 (0.000) | 0.000 (0.000) | 17.220 (1.479) | 0.000 (0.000) |
| 24 | $(1,0,1,1,1,\mathcal{E})$ | 3.365 (1.148) | 0.000 (0.000) | 0.000 (0.000) | 17.217 (1.478) | 0.000 (0.000) |
| 25 | $(1,1,0,0,0,\mathcal{E})$ | 1.755 (1.194) | 0.000 (0.000) | 0.000 (0.000) | 10.641 (0.821) | 0.000 (0.000) |
| 26 | $(1,1,0,0,1,\mathcal{E})$ | 1.804 (1.124) | 0.000 (0.000) | 0.000 (0.000) | 10.641 (0.821) | 0.000 (0.000) |
| 27 | $(1,1,0,1,0,\mathcal{E})$ | 3.732 (0.740) | 0.000 (0.000) | 0.000 (0.000) | 17.220 (1.479) | 0.000 (0.000) |
| 28 | $(1,1,0,1,1,\mathcal{E})$ | 3.732 (0.740) | 0.000 (0.000) | 0.000 (0.000) | 17.220 (1.479) | 0.000 (0.000) |
| 29 | $(1,1,1,0,0,\mathcal{E})$ | 1.755 (1.194) | 0.000 (0.000) | 0.000 (0.000) | 10.641 (0.821) | 0.000 (0.000) |
| 30 | $(1,1,1,0,1,\mathcal{E})$ | 1.755 (1.194) | 0.000 (0.000) | 0.000 (0.000) | 10.641 (0.821) | 0.000 (0.000) |
| 31 | $(1,1,1,1,0,\mathcal{E})$ | 3.692 (0.667) | 0.000 (0.000) | 0.000 (0.000) | 17.220 (1.479) | 0.000 (0.000) |
| 32 | $(1,1,1,1,1,\mathcal{E})$ | 3.692 (0.667) | 0.000 (0.000) | 0.000 (0.000) | 17.220 (1.479) | 0.000 (0.000) |

effect of $X_2$ on $Y$ to be -3.353, i.e., $E(Y_{X_2=1} - Y_{X_2=0}) \approx -3.353$, which suggests that the higher dose lead to weight loss. Figure S2 also provides a simple descriptive statistical analysis of these potential risk factors. We observe that as the toxin level increases, the occurrence rate of organ disease also increases. In addition, males showed higher variability in organ abnormalities compared to females, as demonstrated in previous studies (Bianco et al., 2023). These empirical findings align with monotonicity assumption 3.2, i.e., $X_3(0,0) \leq \{X_3(0,1), X_3(1,0)\} \leq X_3(1,1)$. In this data analysis, we choose not to binarize the outcome and, therefore, do not report the results of the binarized posterior causal effects, as the outcome variable fails to satisfy the monotonicity assumption regarding the causes (Assumption 2(b) in Lu et al. (2023)).

We now present the estimation results of postICEs based on various observed evidence in Table S14. According to Corollary 3.7 and Figure S2, we know that the postICEs are only related to the parent nodes of body weight, namely $X_1$ (gender) and $X_3$ (organ disease). Considering the evidence $(X_1, X_3, \mathcal{E}) = (0, 1, Y < 27)$, we observe PostICE$(Y_{x'} \mid x, Y < 27) = 10.51$, which indicates that changing from male mice without organ disease (i.e., $(x_1', x_3') = (1, 0)$) to female mice with organ disease results in the most significant weight loss, totaling 10.51. For the evidence $(x_1, x_3, \mathcal{E}) = (0, 0, Y < 27)$, we observe PostICE$(Y_{x'} \mid x, Y < 27) = -1.42$, which indicates that changing from female mice with organ disease (i.e., $(x_1', x_3') = (0, 1)$) to female mice without organ disease results in a slight decrease of about 1.42. In summary, for a given posterior evidence, changing from female mice ($X_1 = 0$) to male mice ($X_1 = 1$) results in an increase in body weight,

Table S9: The simulation results for the postNDE under a sample size of 2000.

| Case | Evidence $(x, \mathcal{E})$ | PostNDE$(X_k \Rightarrow Y \mid x, Y > 140)$ | | | | |
|---|---|---|---|---|---|---|
| | | $X_1$ | $X_2$ | $X_3$ | $X_4$ | $X_5$ |
| 1 | $(0,0,0,0,0,\mathcal{E})$ | 1.808 (0.863) | 0.000 (0.000) | 0.000 (0.000) | 3.469 (0.217) | 0.000 (0.000) |
| 2 | $(0,0,0,0,1,\mathcal{E})$ | 1.808 (0.863) | 0.000 (0.000) | 0.000 (0.000) | 3.469 (0.217) | 0.000 (0.000) |
| 3 | $(0,0,0,1,0,\mathcal{E})$ | 5.239 (0.361) | 0.000 (0.000) | 0.000 (0.000) | 13.226 (0.554) | 0.000 (0.000) |
| 4 | $(0,0,0,1,1,\mathcal{E})$ | 5.239 (0.361) | 0.000 (0.000) | 0.000 (0.000) | 13.226 (0.554) | 0.000 (0.000) |
| 5 | $(0,0,1,0,0,\mathcal{E})$ | 1.808 (0.863) | 0.000 (0.000) | 0.914 (0.200) | 3.469 (0.217) | 0.000 (0.000) |
| 6 | $(0,0,1,0,1,\mathcal{E})$ | 1.808 (0.863) | 0.000 (0.000) | 0.000 (0.000) | 3.469 (0.217) | 0.000 (0.000) |
| 7 | $(0,0,0,1,0,\mathcal{E})$ | 5.239 (0.361) | 0.000 (0.000) | 0.000 (0.000) | 13.226 (0.554) | 0.000 (0.000) |
| 8 | $(0,0,1,1,1,\mathcal{E})$ | 5.239 (0.361) | 0.000 (0.000) | 0.000 (0.000) | 13.226 (0.554) | 0.000 (0.000) |
| 9 | $(0,1,0,0,0,\mathcal{E})$ | 1.808 (0.863) | 0.000 (0.000) | 0.000 (0.000) | 3.469 (0.217) | 0.000 (0.000) |
| 10 | $(0,1,0,0,1,\mathcal{E})$ | 1.808 (0.863) | 0.000 (0.000) | 0.000 (0.000) | 3.469 (0.217) | 0.000 (0.000) |
| 11 | $(0,1,0,1,0,\mathcal{E})$ | 5.239 (0.361) | 0.000 (0.000) | 0.000 (0.000) | 13.226 (0.554) | 0.000 (0.000) |
| 12 | $(0,1,0,1,1,\mathcal{E})$ | 5.239 (0.361) | 0.000 (0.000) | 0.000 (0.000) | 13.226 (0.554) | 0.000 (0.000) |
| 13 | $(0,1,1,0,0,\mathcal{E})$ | 1.808 (0.863) | 0.000 (0.000) | 0.000 (0.000) | 3.469 (0.217) | 0.000 (0.000) |
| 14 | $(0,1,1,0,1,\mathcal{E})$ | 1.808 (0.863) | 0.000 (0.000) | 0.000 (0.000) | 3.469 (0.217) | 0.000 (0.000) |
| 15 | $(0,1,1,1,0,\mathcal{E})$ | 5.239 (0.361) | 0.000 (0.000) | 0.000 (0.000) | 13.226 (0.554) | 0.000 (0.000) |
| 16 | $(0,1,1,1,1,\mathcal{E})$ | 5.239 (0.361) | 0.000 (0.000) | 0.000 (0.000) | 13.226 (0.554) | 0.000 (0.000) |
| 17 | $(1,0,0,0,0,\mathcal{E})$ | 1.920 (0.921) | 0.000 (0.000) | 0.000 (0.000) | 10.501 (0.625) | 0.000 (0.000) |
| 18 | $(1,0,0,0,1,\mathcal{E})$ | 1.922 (0.922) | 0.000 (0.000) | 0.000 (0.000) | 10.501 (0.625) | 0.000 (0.000) |
| 19 | $(1,0,0,1,0,\mathcal{E})$ | 3.310 (0.622) | 0.000 (0.000) | 0.000 (0.000) | 17.055 (1.073) | 0.000 (0.000) |
| 20 | $(1,0,0,1,1,\mathcal{E})$ | 3.310 (0.622) | 0.000 (0.000) | 0.000 (0.000) | 17.055 (1.073) | 0.000 (0.000) |
| 21 | $(1,0,1,0,0,\mathcal{E})$ | 1.920 (0.921) | 0.000 (0.000) | 0.000 (0.000) | 10.501 (0.625) | 0.000 (0.000) |
| 22 | $(1,0,1,0,1,\mathcal{E})$ | 1.920 (0.921) | 0.000 (0.000) | 0.000 (0.000) | 10.501 (0.625) | 0.000 (0.000) |
| 23 | $(1,0,1,1,0,\mathcal{E})$ | 3.318 (0.724) | 0.000 (0.000) | 0.000 (0.000) | 17.055 (1.073) | 0.000 (0.000) |
| 24 | $(1,0,1,1,1,\mathcal{E})$ | 3.318 (0.724) | 0.000 (0.000) | 0.000 (0.000) | 17.055 (1.073) | 0.000 (0.000) |
| 25 | $(1,1,0,0,0,\mathcal{E})$ | 1.920 (0.921) | 0.000 (0.000) | 0.000 (0.000) | 10.501 (0.625) | 0.000 (0.000) |
| 26 | $(1,1,0,0,1,\mathcal{E})$ | 1.804 (1.124) | 0.000 (0.000) | 0.000 (0.000) | 10.641 (0.821) | 0.000 (0.000) |
| 27 | $(1,1,0,1,0,\mathcal{E})$ | 3.753 (0.523) | 0.000 (0.000) | 0.000 (0.000) | 17.055 (1.073) | 0.000 (0.000) |
| 28 | $(1,1,0,1,1,\mathcal{E})$ | 3.753 (0.523) | 0.000 (0.000) | 0.000 (0.000) | 17.055 (1.073) | 0.000 (0.000) |
| 29 | $(1,1,1,0,0,\mathcal{E})$ | 1.920 (0.921) | 0.000 (0.000) | 0.000 (0.000) | 10.501 (0.625) | 0.000 (0.000) |
| 30 | $(1,1,1,0,1,\mathcal{E})$ | 1.920 (0.921) | 0.000 (0.000) | 0.000 (0.000) | 10.501 (0.625) | 0.000 (0.000) |
| 31 | $(1,1,1,1,0,\mathcal{E})$ | 3.760 (0.461) | 0.000 (0.000) | 0.000 (0.000) | 17.055 (1.073) | 0.000 (0.000) |
| 32 | $(1,1,1,1,1,\mathcal{E})$ | 3.760 (0.461) | 0.000 (0.000) | 0.000 (0.000) | 17.055 (1.073) | 0.000 (0.000) |

while changing from mice without organ disease ($X_3 = 0$) to mice with organ disease ($X_3 = 1$) results in a decrease in body weight in most cases.

Table S15 presents the estimation results of the posterior causal estimands for each potential risk factor. Given the evidence $(X_1, X_2, X_3, \mathcal{E}) = (1, 1, 1, Y < 27)$, we find that the PostNDE value for gender $X_1$ is the highest, indicating that gender might be the most important direct factor, followed by organ disease $X_3$. The PostNDE value for toxin dose $X_2$ is 0, which is consistent with the conclusions drawn from the Bayesian network in Figure S2. In addition, given the evidence $(X_1, X_2, X_3, \mathcal{E}) = (1, 1, 1, Y < 27)$, we observed that both gender $X_1$ and toxin dose $X_2$ have an indirect effect on the outcome, with the absolute value of PostNIE for gender $X_1$ being significantly greater than that for toxin dose $X_2$, while organ disease $X_3$ has no indirect effect, suggesting that gender may be the most important indirect factor influencing body weight among the three factors, followed by toxin dose $X_2$. However, in terms of the postTCE, organ disease $X_3$ has the largest value, suggesting that organ disease $X_3$ is the most important total risk factor for low body weight, followed by toxin dose $X_2$, while gender $X_1$ has the least effect.

Under the evidence $(1, 0, 0, Y < 27)$ and $(0, 1, 1, Y < 27)$, toxin dose $X_2$ exhibits the most significant PostNIE value for weight loss, indicating that toxin dose $X_2$ is the most important indirect risk factor in these situations. In summary, for most evidence in Table S15, either gender $X_1$ or organ disease $X_3$ has the highest absolute PostTCE value, especially $X_3$,

Table S10: The simulation results for the postNDE under a sample size of 10000.

| Case | Evidence $(x, \mathcal{E})$ | postNDE($X_k \Rightarrow Y \mid x, Y > 140$) | | | | |
|---|---|---|---|---|---|---|
| | | $X_1$ | $X_2$ | $X_3$ | $X_4$ | $X_5$ |
| 1 | $(0,0,0,0,0,\mathcal{E})$ | 1.961(0.351) | 0.000(0.000) | 0.000(0.000) | 3.504(0.102) | 0.000(0.000) |
| 2 | $(0,0,0,0,1,\mathcal{E})$ | 1.961(0.351) | 0.000(0.000) | 0.000(0.000) | 3.504(0.102) | 0.000(0.000) |
| 3 | $(0,0,0,1,0,\mathcal{E})$ | 5.259(0.171) | 0.000(0.000) | 0.000(0.000) | 13.259(0.261) | 0.000(0.000) |
| 4 | $(0,0,0,1,1,\mathcal{E})$ | 5.259(0.171) | 0.000(0.000) | 0.000(0.000) | 13.259(0.261) | 0.000(0.000) |
| 5 | $(0,0,1,0,0,\mathcal{E})$ | 1.961(0.351) | 0.000(0.000) | 0.000(0.000) | 3.504(0.102) | 0.000(0.000) |
| 6 | $(0,0,1,0,1,\mathcal{E})$ | 1.961(0.351) | 0.000(0.000) | 0.000(0.000) | 3.504(0.102) | 0.000(0.000) |
| 7 | $(0,0,0,1,0,\mathcal{E})$ | 5.259(0.171) | 0.000(0.000) | 0.000(0.000) | 13.259(0.261) | 0.000(0.000) |
| 8 | $(0,0,1,1,1,\mathcal{E})$ | 5.259(0.171) | 0.000(0.000) | 0.000(0.000) | 13.259(0.261) | 0.000(0.000) |
| 9 | $(0,1,0,0,0,\mathcal{E})$ | 1.961(0.351) | 0.000(0.000) | 0.000(0.000) | 3.504(0.102) | 0.000(0.000) |
| 10 | $(0,1,0,0,1,\mathcal{E})$ | 1.961(0.351) | 0.000(0.000) | 0.000(0.000) | 3.504(0.102) | 0.000(0.000) |
| 11 | $(0,1,0,1,0,\mathcal{E})$ | 5.259(0.171) | 0.000(0.000) | 0.000(0.000) | 13.259(0.261) | 0.000(0.000) |
| 12 | $(0,1,0,1,1,\mathcal{E})$ | 5.259(0.171) | 0.000(0.000) | 0.000(0.000) | 13.259(0.261) | 0.000(0.000) |
| 13 | $(0,1,1,0,0,\mathcal{E})$ | 1.961(0.351) | 0.000(0.000) | 0.000(0.000) | 3.504(0.102) | 0.000(0.000) |
| 14 | $(0,1,1,0,1,\mathcal{E})$ | 1.961(0.351) | 0.000(0.000) | 0.000(0.000) | 3.504(0.102) | 0.000(0.000) |
| 15 | $(0,1,1,1,0,\mathcal{E})$ | 5.259(0.171) | 0.000(0.000) | 0.000(0.000) | 13.259(0.261) | 0.000(0.000) |
| 16 | $(0,1,1,1,1,\mathcal{E})$ | 5.259(0.171) | 0.000(0.000) | 0.000(0.000) | 13.259(0.261) | 0.000(0.000) |
| 17 | $(1,0,0,0,0,\mathcal{E})$ | 1.983(0.382) | 0.000(0.000) | 0.000(0.000) | 10.513(0.267) | 0.000(0.000) |
| 18 | $(1,0,0,0,1,\mathcal{E})$ | 1.983(0.382) | 0.000(0.000) | 0.000(0.000) | 10.513(0.267) | 0.000(0.000) |
| 19 | $(1,0,0,1,0,\mathcal{E})$ | 3.357(0.267) | 0.000(0.000) | 0.000(0.000) | 17.050(0.466) | 0.000(0.000) |
| 20 | $(1,0,0,1,1,\mathcal{E})$ | 3.357(0.267) | 0.000(0.000) | 0.000(0.000) | 17.050(0.466) | 0.000(0.000) |
| 21 | $(1,0,1,0,0,\mathcal{E})$ | 1.983(0.382) | 0.000(0.000) | 0.000(0.000) | 10.513(0.267) | 0.000(0.000) |
| 22 | $(1,0,1,0,1,\mathcal{E})$ | 1.983(0.382) | 0.000(0.000) | 0.000(0.000) | 10.513(0.267) | 0.000(0.000) |
| 23 | $(1,0,1,1,0,\mathcal{E})$ | 3.364(0.322) | 0.000(0.000) | 0.000(0.000) | 17.050(0.466) | 0.000(0.000) |
| 24 | $(1,0,1,1,1,\mathcal{E})$ | 3.364(0.322) | 0.000(0.000) | 0.000(0.000) | 17.050(0.466) | 0.000(0.000) |
| 25 | $(1,1,0,0,0,\mathcal{E})$ | 1.983(0.382) | 0.000(0.000) | 0.000(0.000) | 10.513(0.267) | 0.000(0.000) |
| 26 | $(1,1,0,0,1,\mathcal{E})$ | 1.983(0.382) | 0.000(0.000) | 0.000(0.000) | 10.513(0.267) | 0.000(0.000) |
| 27 | $(1,1,0,1,0,\mathcal{E})$ | 3.796(0.229) | 0.000(0.000) | 0.000(0.000) | 17.050(0.466) | 0.000(0.000) |
| 28 | $(1,1,0,1,1,\mathcal{E})$ | 3.798(0.228) | 0.000(0.000) | 0.000(0.000) | 17.045(0.466) | 0.000(0.000) |
| 29 | $(1,1,1,0,0,\mathcal{E})$ | 1.983(0.382) | 0.000(0.000) | 0.000(0.000) | 10.513(0.267) | 0.000(0.000) |
| 30 | $(1,1,1,0,1,\mathcal{E})$ | 1.978(0.381) | 0.000(0.000) | 0.000(0.000) | 10.519(0.269) | 0.000(0.000) |
| 31 | $(1,1,1,1,0,\mathcal{E})$ | 3.798(0.228) | 0.000(0.000) | 0.000(0.000) | 17.047(0.470) | 0.000(0.000) |
| 32 | $(1,1,1,1,1,\mathcal{E})$ | 3.790(0.195) | 0.000(0.000) | 0.000(0.000) | 17.076(0.465) | 0.000(0.000) |

suggesting that organ disease is the most important risk factor leading to weight loss.

Table S11: The simulation results for the postNIE under a sample size of 1000.

| Case | Evidence $(x, \mathcal{E})$ | PostNIE$(X_k \Rightarrow Y \mid x, Y > 140)$ | | | | |
|---|---|---|---|---|---|---|
| | | $X_1$ | $X_2$ | $X_3$ | $X_4$ | $X_5$ |
| 1 | $(0,0,0,0,0,\mathcal{E})$ | 4.157 (1.094) | 0.926 (0.186) | -0.001 (0.376) | 0.000 (0.000) | 0.000 (0.000) |
| 2 | $(0,0,0,0,1,\mathcal{E})$ | 4.159 (1.093) | 0.926 (0.186) | -0.001 (0.376) | 0.000 (0.000) | 0.000 (0.000) |
| 3 | $(0,0,0,1,0,\mathcal{E})$ | 0.000 (0.000) | 0.000 (0.000) | 0.000 (0.000) | 0.000 (0.000) | 0.000 (0.000) |
| 4 | $(0,0,0,1,1,\mathcal{E})$ | 0.000 (0.000) | 0.000 (0.000) | 0.000 (0.000) | 0.000 (0.000) | 0.000 (0.000) |
| 5 | $(0,0,1,0,0,\mathcal{E})$ | 4.058 (1.995) | 0.903 (0.286) | 0.000 (0.000) | 0.000 (0.000) | 0.000 (0.000) |
| 6 | $(0,0,1,0,1,\mathcal{E})$ | 4.082 (2.013) | 0.901 (0.286) | 0.000 (0.000) | 0.000 (0.000) | 0.000 (0.000) |
| 7 | $(0,0,0,1,0,\mathcal{E})$ | 0.000 (0.000) | 0.000 (0.000) | -1.186 (5.424) | 0.000 (0.000) | 0.000 (0.000) |
| 8 | $(0,0,1,1,1,\mathcal{E})$ | 0.000 (0.000) | 0.000 (0.000) | -1.186 (5.424) | 0.000 (0.000) | 0.000 (0.000) |
| 9 | $(0,1,0,0,0,\mathcal{E})$ | 5.580 (1.742) | 0.000 (0.000) | -0.056 (0.429) | 0.000 (0.000) | 0.000 (0.000) |
| 10 | $(0,1,0,0,1,\mathcal{E})$ | 5.580 (1.742) | 0.000 (0.000) | -0.056 (0.429) | 0.000 (0.000) | 0.000 (0.000) |
| 11 | $(0,1,0,1,0,\mathcal{E})$ | 0.000 (0.000) | 5.801 (1.560) | 0.000 (0.000) | 0.000 (0.000) | 0.000 (0.000) |
| 12 | $(0,1,0,1,1,\mathcal{E})$ | 0.000 (0.000) | 5.801 (1.560) | 0.000 (0.000) | 0.000 (0.000) | 0.000 (0.000) |
| 13 | $(0,1,1,0,0,\mathcal{E})$ | 5.707 (0.874) | 0.000 (0.000) | 0.000 (0.000) | 0.000 (0.000) | 0.000 (0.000) |
| 14 | $(0,1,1,0,1,\mathcal{E})$ | 5.707 (0.874) | 0.000 (0.000) | 0.000 (0.000) | 0.000 (0.000) | 0.000 (0.000) |
| 15 | $(0,1,1,1,0,\mathcal{E})$ | 0.000 (0.000) | 5.870 (1.056) | -0.094 (1.876) | 0.000 (0.000) | 0.000 (0.000) |
| 16 | $(0,1,1,1,1,\mathcal{E})$ | 0.000 (0.000) | 5.870 (1.056) | -0.094 (1.876) | 0.000 (0.000) | 0.000 (0.000) |
| 17 | $(1,0,0,0,0,\mathcal{E})$ | 0.000 (0.000) | 4.616 (1.206) | -0.266 (3.567) | 0.000 (0.000) | 0.000 (0.000) |
| 18 | $(1,0,0,0,1,\mathcal{E})$ | 0.000 (0.000) | 4.616 (1.206) | -0.246 (3.591) | 0.000 (0.000) | 0.000 (0.000) |
| 19 | $(1,0,0,1,0,\mathcal{E})$ | 9.322 (1.777) | 0.000 (0.000) | 0.000 (0.000) | 0.000 (0.000) | 0.000 (0.000) |
| 20 | $(1,0,0,1,1,\mathcal{E})$ | 9.322 (1.777) | 0.000 (0.000) | 0.000 (0.000) | 0.000 (0.000) | 0.000 (0.000) |
| 21 | $(1,0,1,0,0,\mathcal{E})$ | 0.000 (0.000) | 4.544 (2.096) | 0.000 (0.000) | 0.000 (0.000) | 0.000 (0.000) |
| 22 | $(1,0,1,0,1,\mathcal{E})$ | 0.000 (0.000) | 4.159 (2.725) | 0.000 (0.000) | 0.000 (0.000) | 0.000 (0.000) |
| 23 | $(1,0,1,1,0,\mathcal{E})$ | 8.688 (4.183) | 0.000 (0.000) | -1.352 (6.608) | 0.000 (0.000) | 0.000 (0.000) |
| 24 | $(1,0,1,1,1,\mathcal{E})$ | 8.708 (4.163) | 0.000 (0.000) | -1.345 (6.613) | 0.000 (0.000) | 0.000 (0.000) |
| 25 | $(1,1,0,0,0,\mathcal{E})$ | 0.000 (0.000) | 0.000 (0.000) | -1.177 (5.164) | 0.000 (0.000) | 0.000 (0.000) |
| 26 | $(1,1,0,0,1,\mathcal{E})$ | 0.000 (0.000) | 0.000 (0.000) | -1.177 (5.164) | 0.000 (0.000) | 0.000 (0.000) |
| 27 | $(1,1,0,1,0,\mathcal{E})$ | 6.711 (1.876) | 4.408 (2.022) | 0.000 (0.000) | 0.000 (0.000) | 0.000 (0.000) |
| 28 | $(1,1,0,1,1,\mathcal{E})$ | 6.711 (1.876) | 4.408 (2.022) | 0.000 (0.000) | 0.000 (0.000) | 0.000 (0.000) |
| 29 | $(1,1,1,0,0,\mathcal{E})$ | 0.000 (0.000) | 0.000 (0.000) | 0.000 (0.000) | 0.000 (0.000) | 0.000 (0.000) |
| 30 | $(1,1,1,0,1,\mathcal{E})$ | 0.000 (0.000) | 0.000 (0.000) | 0.000 (0.000) | 0.000 (0.000) | 0.000 (0.000) |
| 31 | $(1,1,1,1,0,\mathcal{E})$ | 6.902 (0.902) | 4.582 (1.508) | 0.043 (1.996) | 0.000 (0.000) | 0.000 (0.000) |
| 32 | $(1,1,1,1,1,\mathcal{E})$ | 6.902 (0.902) | 4.588 (1.503) | 0.043 (1.996) | 0.000 (0.000) | 0.000 (0.000) |

Table S12: The simulation results for the postNIE under a sample size of 2000.

| Case | Evidence $(x, \mathcal{E})$ | PostNIE$(X_k \Rightarrow Y \mid x, Y > 140)$ | | | | |
|---|---|---|---|---|---|---|
| | | $X_1$ | $X_2$ | $X_3$ | $X_4$ | $X_5$ |
| 1 | $(0,0,0,0,0,\mathcal{E})$ | 4.059 (0.748) | 0.921 (0.120) | -0.004 (0.256) | 0.000 (0.000) | 0.000 (0.000) |
| 2 | $(0,0,0,0,1,\mathcal{E})$ | 4.059 (0.748) | 0.921 (0.120) | -0.004 (0.256) | 0.000 (0.000) | 0.000 (0.000) |
| 3 | $(0,0,0,1,0,\mathcal{E})$ | 0.000 (0.000) | 0.000 (0.000) | 0.000 (0.000) | 0.000 (0.000) | 0.000 (0.000) |
| 4 | $(0,0,0,1,1,\mathcal{E})$ | 0.000 (0.000) | 0.000 (0.000) | 0.000 (0.000) | 0.000 (0.000) | 0.000 (0.000) |
| 5 | $(0,0,1,0,0,\mathcal{E})$ | 4.120 (1.418) | 0.914 (0.200) | 0.000 (0.000) | 0.000 (0.000) | 0.000 (0.000) |
| 6 | $(0,0,1,0,1,\mathcal{E})$ | 4.120 (1.418) | 0.914 (0.200) | 0.000 (0.000) | 0.000 (0.000) | 0.000 (0.000) |
| 7 | $(0,0,0,1,0,\mathcal{E})$ | 0.000 (0.000) | 0.000 (0.000) | -0.576 (3.295) | 0.000 (0.000) | 0.000 (0.000) |
| 8 | $(0,0,1,1,1,\mathcal{E})$ | 0.000 (0.000) | 0.000 (0.000) | -0.576 (3.295) | 0.000 (0.000) | 0.000 (0.000) |
| 9 | $(0,1,0,0,0,\mathcal{E})$ | 5.531 (1.246) | 0.000 (0.000) | -0.005 (0.271) | 0.000 (0.000) | 0.000 (0.000) |
| 10 | $(0,1,0,0,1,\mathcal{E})$ | 5.531 (1.246) | 0.000 (0.000) | -0.005 (0.271) | 0.000 (0.000) | 0.000 (0.000) |
| 11 | $(0,1,0,1,0,\mathcal{E})$ | 0.000 (0.000) | 5.808 (1.006) | 0.000 (0.000) | 0.000 (0.000) | 0.000 (0.000) |
| 12 | $(0,1,0,1,1,\mathcal{E})$ | 0.000 (0.000) | 5.808 (1.006) | 0.000 (0.000) | 0.000 (0.000) | 0.000 (0.000) |
| 13 | $(0,1,1,0,0,\mathcal{E})$ | 5.544 (0.653) | 0.000 (0.000) | 0.000 (0.000) | 0.000 (0.000) | 0.000 (0.000) |
| 14 | $(0,1,1,0,1,\mathcal{E})$ | 5.544 (0.653) | 0.000 (0.000) | 0.000 (0.000) | 0.000 (0.000) | 0.000 (0.000) |
| 15 | $(0,1,1,1,0,\mathcal{E})$ | 0.000 (0.000) | 5.887 (0.746) | 0.041 (1.254) | 0.000 (0.000) | 0.000 (0.000) |
| 16 | $(0,1,1,1,1,\mathcal{E})$ | 0.000 (0.000) | 5.887 (0.746) | 0.041 (1.254) | 0.000 (0.000) | 0.000 (0.000) |
| 17 | $(1,0,0,0,0,\mathcal{E})$ | 0.000 (0.000) | 4.606 (0.828) | 0.025 (2.427) | 0.000 (0.000) | 0.000 (0.000) |
| 18 | $(1,0,0,0,1,\mathcal{E})$ | 0.000 (0.000) | 4.606 (0.828) | 0.025 (2.427) | 0.000 (0.000) | 0.000 (0.000) |
| 19 | $(1,0,0,1,0,\mathcal{E})$ | 9.308 (1.114) | 0.000 (0.000) | 0.000 (0.000) | 0.000 (0.000) | 0.000 (0.000) |
| 20 | $(1,0,0,1,1,\mathcal{E})$ | 9.308 (1.114) | 0.000 (0.000) | 0.000 (0.000) | 0.000 (0.000) | 0.000 (0.000) |
| 21 | $(1,0,1,0,0,\mathcal{E})$ | 0.000 (0.000) | 4.406 (1.524) | 0.000 (0.000) | 0.000 (0.000) | 0.000 (0.000) |
| 22 | $(1,0,1,0,1,\mathcal{E})$ | 0.000 (0.000) | 4.296 (1.620) | 0.000 (0.000) | 0.000 (0.000) | 0.000 (0.000) |
| 23 | $(1,0,1,1,0,\mathcal{E})$ | 9.227 (2.246) | 0.000 (0.000) | -0.376 (3.501) | 0.000 (0.000) | 0.000 (0.000) |
| 24 | $(1,0,1,1,1,\mathcal{E})$ | 9.227 (2.246) | 0.000 (0.000) | -0.376 (3.501) | 0.000 (0.000) | 0.000 (0.000) |
| 25 | $(1,1,0,0,0,\mathcal{E})$ | 0.000 (0.000) | 0.000 (0.000) | -0.546 (2.909) | 0.000 (0.000) | 0.000 (0.000) |
| 26 | $(1,1,0,0,1,\mathcal{E})$ | 0.000 (0.000) | 0.000 (0.000) | -0.546 (2.909) | 0.000 (0.000) | 0.000 (0.000) |
| 27 | $(1,1,0,1,0,\mathcal{E})$ | 6.819 (1.306) | 4.437 (1.473) | 0.000 (0.000) | 0.000 (0.000) | 0.000 (0.000) |
| 28 | $(1,1,0,1,1,\mathcal{E})$ | 6.819 (1.306) | 4.437 (1.473) | 0.000 (0.000) | 0.000 (0.000) | 0.000 (0.000) |
| 29 | $(1,1,1,0,0,\mathcal{E})$ | 0.000 (0.000) | 0.000 (0.000) | 0.000 (0.000) | 0.000 (0.000) | 0.000 (0.000) |
| 30 | $(1,1,1,0,1,\mathcal{E})$ | 0.000 (0.000) | 0.000 (0.000) | 0.000 (0.000) | 0.000 (0.000) | 0.000 (0.000) |
| 31 | $(1,1,1,1,0,\mathcal{E})$ | 6.827 (0.693) | 4.489 (1.034) | 0.007 (1.395) | 0.000 (0.000) | 0.000 (0.000) |
| 32 | $(1,1,1,1,1,\mathcal{E})$ | 6.827 (0.693) | 4.489 (1.034) | 0.007 (1.395) | 0.000 (0.000) | 0.000 (0.000) |

Table S13: The simulation results for postNIE under sample size 10000.

| Case | Evidence $(x, \mathcal{E})$ | PostNIE$(X_k \Rightarrow Y \mid x, Y > 140)$ | | | | |
|------|------|------|------|------|------|------|
| | | $X_1$ | $X_2$ | $X_3$ | $X_4$ | $X_5$ |
| 1 | $(0,0,0,0,0,\mathcal{E})$ | 4.004 (0.331) | 0.934 (0.057) | 0.001 (0.122) | 0.000 (0.000) | 0.000 (0.000) |
| 2 | $(0,0,0,0,1,\mathcal{E})$ | 4.004 (0.331) | 0.934 (0.057) | 0.001 (0.122) | 0.000 (0.000) | 0.000 (0.000) |
| 3 | $(0,0,0,1,0,\mathcal{E})$ | 0.000 (0.000) | 0.000 (0.000) | 0.000 (0.000) | 0.000 (0.000) | 0.000 (0.000) |
| 4 | $(0,0,0,1,1,\mathcal{E})$ | 0.000 (0.000) | 0.000 (0.000) | 0.000 (0.000) | 0.000 (0.000) | 0.000 (0.000) |
| 5 | $(0,0,1,0,0,\mathcal{E})$ | 3.989 (0.590) | 0.931 (0.088) | 0.000 (0.000) | 0.000 (0.000) | 0.000 (0.000) |
| 6 | $(0,0,1,0,1,\mathcal{E})$ | 3.989 (0.590) | 0.931 (0.088) | 0.000 (0.000) | 0.000 (0.000) | 0.000 (0.000) |
| 7 | $(0,0,0,1,0,\mathcal{E})$ | 0.000 (0.000) | 0.000 (0.000) | -0.098 (1.409) | 0.000 (0.000) | 0.000 (0.000) |
| 8 | $(0,0,1,1,1,\mathcal{E})$ | 0.000 (0.000) | 0.000 (0.000) | -0.098 (1.409) | 0.000 (0.000) | 0.000 (0.000) |
| 9 | $(0,1,0,0,0,\mathcal{E})$ | 5.458 (0.491) | 0.000 (0.000) | -0.004 (0.256) | 0.000 (0.000) | 0.000 (0.000) |
| 10 | $(0,1,0,0,1,\mathcal{E})$ | 5.458 (0.491) | 0.000 (0.000) | -0.004 (0.126) | 0.000 (0.000) | 0.000 (0.000) |
| 11 | $(0,1,0,1,0,\mathcal{E})$ | 0.000 (0.000) | 5.880 (0.466) | 0.000 (0.000) | 0.000 (0.000) | 0.000 (0.000) |
| 12 | $(0,1,0,1,1,\mathcal{E})$ | 0.000 (0.000) | 5.880 (0.466) | 0.000 (0.000) | 0.000 (0.000) | 0.000 (0.000) |
| 13 | $(0,1,1,0,0,\mathcal{E})$ | 5.472 (0.259) | 0.000 (0.000) | 0.000 (0.000) | 0.000 (0.000) | 0.000 (0.000) |
| 14 | $(0,1,1,0,1,\mathcal{E})$ | 5.472 (0.259) | 0.000 (0.000) | 0.000 (0.000) | 0.000 (0.000) | 0.000 (0.000) |
| 15 | $(0,1,1,1,0,\mathcal{E})$ | 0.000 (0.000) | 5.886 (0.336) | -0.005 (0.577) | 0.000 (0.000) | 0.000 (0.000) |
| 16 | $(0,1,1,1,1,\mathcal{E})$ | 0.000 (0.000) | 5.886 (0.336) | -0.005 (0.577) | 0.000 (0.000) | 0.000 (0.000) |
| 17 | $(1,0,0,0,0,\mathcal{E})$ | 0.000 (0.000) | 4.664 (0.346) | -0.037 (1.038) | 0.000 (0.000) | 0.000 (0.000) |
| 18 | $(1,0,0,0,1,\mathcal{E})$ | 0.000 (0.000) | 4.664 (0.346) | -0.037 (1.038) | 0.000 (0.000) | 0.000 (0.000) |
| 19 | $(1,0,0,1,0,\mathcal{E})$ | 9.275 (0.501) | 0.000 (0.000) | 0.000 (0.000) | 0.000 (0.000) | 0.000 (0.000) |
| 20 | $(1,0,0,1,1,\mathcal{E})$ | 9.275 (0.501) | 0.000 (0.000) | 0.000 (0.000) | 0.000 (0.000) | 0.000 (0.000) |
| 21 | $(1,0,1,0,0,\mathcal{E})$ | 0.000 (0.000) | 4.640 (0.586) | 0.000 (0.000) | 0.000 (0.000) | 0.000 (0.000) |
| 22 | $(1,0,1,0,1,\mathcal{E})$ | 0.000 (0.000) | 4.640 (0.586) | 0.000 (0.000) | 0.000 (0.000) | 0.000 (0.000) |
| 23 | $(1,0,1,1,0,\mathcal{E})$ | 9.220 (1.009) | 0.000 (0.000) | -0.114 (1.414) | 0.000 (0.000) | 0.000 (0.000) |
| 24 | $(1,0,1,1,1,\mathcal{E})$ | 9.220 (1.009) | 0.000 (0.000) | -0.114 (1.414) | 0.000 (0.000) | 0.000 (0.000) |
| 25 | $(1,1,0,0,0,\mathcal{E})$ | 0.000 (0.000) | 0.000 (0.000) | -0.069 (1.093) | 0.000 (0.000) | 0.000 (0.000) |
| 26 | $(1,1,0,0,1,\mathcal{E})$ | 0.000 (0.000) | 0.000 (0.000) | -0.069 (1.093) | 0.000 (0.000) | 0.000 (0.000) |
| 27 | $(1,1,0,1,0,\mathcal{E})$ | 6.800 (0.551) | 4.541 (0.606) | 0.000 (0.000) | 0.000 (0.000) | 0.000 (0.000) |
| 28 | $(1,1,0,1,1,\mathcal{E})$ | 6.793 (0.553) | 4.532 (0.600) | 0.000 (0.000) | 0.000 (0.000) | 0.000 (0.000) |
| 29 | $(1,1,1,0,0,\mathcal{E})$ | 0.000 (0.000) | 0.000 (0.000) | 0.000 (0.000) | 0.000 (0.000) | 0.000 (0.000) |
| 30 | $(1,1,1,0,1,\mathcal{E})$ | 0.000 (0.000) | 0.000 (0.000) | 0.000 (0.000) | 0.000 (0.000) | 0.000 (0.000) |
| 31 | $(1,1,1,1,0,\mathcal{E})$ | 6.812 (0.282) | 4.567 (0.451) | 0.019 (0.577) | 0.000 (0.000) | 0.000 (0.000) |
| 32 | $(1,1,1,1,1,\mathcal{E})$ | 6.816 (0.270) | 4.620 (0.461) | 0.008 (0.584) | 0.000 (0.000) | 0.000 (0.000) |

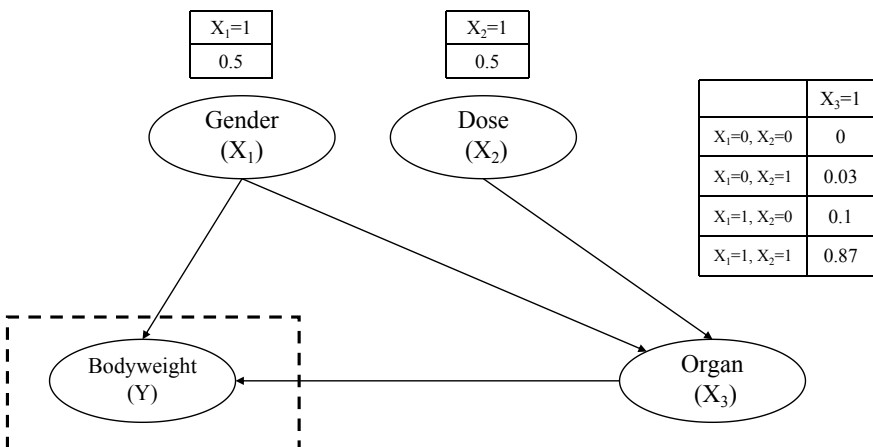

Figure S2: A causal network representing developmental toxicology experiments, including body weight and its potential risk factors.

Table S14: Results of postICEs based on different evidence.

| PostICE($Y_{x'} \mid x, Y < 27$) | $(x_1, x_3) = (0,0)$ | $(x_1, x_3) = (0,1)$ | $(x_1, x_3) = (1,0)$ | $(x_1, x_3) = (1,1)$ |
|---|---|---|---|---|
| $(x_1', x_3') = (0,0)$ | 0.00 | 1.59 | -7.10 | -2.77 |
| $(x_1', x_3') = (0,1)$ | -1.42 | 0.00 | -5.50 | -3.35 |
| $(x_1', x_3') = (1,0)$ | 8.81 | 10.51 | 0.00 | 5.58 |
| $(x_1', x_3') = (1,1)$ | 2.84 | 4.46 | -8.90 | 0.00 |

Table S15: Results of posterior causal estimands based on the various evidence for the NTP dataset.

| Posterior causal estimands | Evidence $X = x$ | | | | | | | |
|---|---|---|---|---|---|---|---|---|
| | $(0,0,0)$ | $(1,0,0)$ | $(0,1,0)$ | $(1,1,0)$ | $(0,0,1)$ | $(1,0,1)$ | $(0,1,1)$ | $(1,1,1)$ |
| PostNDE($X_1 \Rightarrow Y \mid x, Y < 27$) | 8.83 | 7.00 | 8.83 | 7.00 | 10.00 | 8.31 | 10.00 | 8.12 |
| PostNDE($X_2 \Rightarrow Y \mid x, Y < 27$) | 0.00 | 0.00 | 0.00 | 0.00 | 0.00 | 0.00 | 0.00 | 0.00 |
| PostNDE($X_3 \Rightarrow Y \mid x, Y < 27$) | -1.43 | -8.90 | -1.43 | -8.90 | -6.30 | -5.51 | -6.30 | -5.51 |
| PostNIE($X_1 \Rightarrow Y \mid x, Y < 27$) | -0.60 | 0.00 | -5.18 | 0.00 | 0.00 | -5.51 | 0.00 | -5.30 |
| PostNIE($X_2 \Rightarrow Y \mid x, Y < 27$) | -0.05 | -7.58 | 0.00 | 0.00 | 0.00 | 0.00 | -6.30 | -4.87 |
| PostNIE($X_3 \Rightarrow Y \mid x, Y < 27$) | 0.00 | 0.00 | 0.00 | 0.00 | 0.00 | 0.00 | 0.00 | 0.00 |
| PostTCE($X_1 \Rightarrow Y \mid x, Y < 27$) | 8.23 | 7.00 | 3.65 | 7.00 | 10.00 | 2.80 | 10.00 | 2.82 |
| PostTCE($X_2 \Rightarrow Y \mid x, Y < 27$) | -0.05 | -7.58 | 0.00 | 0.00 | 0.00 | 0.00 | -6.30 | -4.87 |
| PostTCE($X_3 \Rightarrow Y \mid x, Y < 27$) | -1.43 | -8.90 | -1.43 | -8.90 | -6.30 | -5.51 | -6.30 | -5.51 |

