# OpenReview forum: "Causal Attribution Analysis for Continuous Outcomes"
_ICML.cc/2025/Conference — ICML 2025 spotlightposter_

### Official Review · Reviewer_ZTX9 · 2025-02-13

**Overall Recommendation:** 5

**Summary:**

Previous studies have focused on attribution problems for binary outcomes, but binarizing continuous outcomes can lead to information loss or bias. To address this, the study introduces posterior causal estimands for evaluating multiple correlated causes in continuous outcomes. These estimands include posterior intervention effects, posterior total causal effects, and posterior natural direct effects. Under assumptions like sequential ignorability, monotonicity, and perfect positive rank, the study establishes the identifiability of these estimands and derives corresponding identification equations. A simple yet effective estimation procedure is proposed, with theoretical guarantees on asymptotic properties. The method is demonstrated through an artificial hypertension example and a real developmental toxicity dataset.

**Claims And Evidence:**

Yes, the claims are very vlear.

**Essential References Not Discussed:**

No.

**Ethical Review Concerns:**

No.

**Experimental Designs Or Analyses:**

It is advisable to include confidence intervals.

**Methods And Evaluation Criteria:**

Yes.

**Other Comments Or Suggestions:**

You have examined posterior natural direct and indirect effects for the variables (X_{k+1},...,X_p). Can you also identify the path‐specific effect, as described by Pearl (2001)? (May be future work)

Judea Pearl. 2001. Direct and indirect effects. In Proceedings of the Seventeenth conference on Uncertainty in artificial intelligence (UAI'01). Morgan Kaufmann Publishers Inc., San Francisco, CA, USA, 411–420.

**Other Strengths And Weaknesses:**

This paper is well-structured. They address a naturally arising causal question that researchers have thus far not explored.

**Questions For Authors:**

1. In line 165, you wrote, “However, this monotonic relationship may not be applicable when the outcome variable is continuous.” However, I believe that the monotonicity assumption is indeed directly applicable to continuous outcomes. Are you suggesting that posterior causal estimands cannot be identified through this monotonicity assumption?

2. Does Assumption 3.3 imply Assumption 3.2? It seems Assumption 3.3 imposes strict monotonicity, whereas Assumption 3.2 only requires non-strict monotonicity.

3. Do you requires all Assumptions 3.1, 3.2, and 3.3 for identifying postNDE, postNIE, and postTCE? The statement Theorem 3.6 may look that postNDE, postNIE, and postTCE are idenrtified from Assumptions 3.1, 3.2.

4. I would like additional clarification regarding Lemma 4.2. Specifically, what does each property guarantee?

5. Do you presume that the probability of the evidence $pr(X=x,{\cal E})\ne 0$?

**Relation To Broader Scientific Literature:**

As the authors indicate, their findings have broad applications in social science, health risk assessment, legal contexts, and explainable AI. They showed in Introduction section.

**Theoretical Claims:**

I have checked the correctness of theoretical claims. They are all correct.

---

> ### Author Rebuttal · Authors · 2025-03-29
>
> ---
> We sincerely thank the reviewer for the thoughtful feedback and strong accept recommendation. Your recognition of our contribution is very encouraging, and your suggestions are highly valuable for improving the paper.
>
>
>
> ---
> **Q1: Path-Specific Effects and Pearl (2001)**
>
> **A1:** Thank you for the interesting comments. Following your suggestion, we revisited Pearl (2001) and examined how path-specific effects can be incorporated into our framework.
>
>
> Given a causal graph $G$ over $(X, Y)$ and a subgraph $l \subseteq G$ representing the path(s) of interest, our goal is to isolate the effect of $X_k$ on $Y$ transmitted specifically through path $l$. Let $\mathrm{Pa}_j$ denote the set of parent nodes of $X_j$, which we decompose as $\mathrm{Pa}_j = \mathrm{Pa}_j(l) \cup \mathrm{Pa}_j(\bar{l})$, where $\mathrm{Pa}_j(l)$ includes parents along path $l$, and $\mathrm{Pa}_j(\bar{l})$ includes parents outside of it. To activate only path $l$, we fix the non-path parents $\mathrm{Pa}_j(\bar{l})$ to their counterfactual values under the reference setting $X_k = 0.$
>
>
> The $l$-specific counterfactual outcome is defined as $Y^l_{X_k = x_k} = (X_{p+1})^l_{X_k = x_k}$, computed under the modified system with only path $l$ active. When $X_k = 0$, this reduces to the usual counterfactual: $Y^l_{X_k = 0} = Y_{X_k = 0}$, since all variables follow the reference path.
> We define the posterior path-specific effect as
> $$
> \mathrm{postPSE}(X_k \Rightarrow Y; x, \mathcal{E}, l) = E\left( Y^l_{X_k = 1} - Y_{X_k= 0} \mid x, \mathcal{E}\right).
> $$
>
> The second term is identifiable via Theorem 3.6. Identifying the first term requires a finer strategy that accounts for node-level structures along path $l$ and their interaction with the evidence $ \mathcal{E} $. We leave this technical development for future work.
>
>
> ---
>
> **Q2: Monotonicity with Continuous Outcomes**
>
> **A2:**  Thank you for the insightful question. Our intention was not to suggest that monotonicity is inappropriate for continuous outcomes, but rather to note that a strong condition such as $Y_x\leq Y_{x'}$ for all $x \preceq x'$ may be restrictive in certain settings. To illustrate this, consider a linear model:
> $$Y = \alpha_0 + \alpha_1 X_1 + \alpha_2 X_2 + \cdots + \alpha_p X_p + \epsilon.$$
> If we assume $Y_x \leq Y_{x'}$ whenever $x \preceq x'$, then this implies that increasing any component of exposure vairables $x$ (with others held fixed or increased) must not decrease $Y$. In a linear model, this requirement implies $\alpha_j \geq 0$ for all $j = 1, \ldots, p$, which may limit model flexibility.
>
> We hence adopt the perfect rank assumption as a more general alternative. It includes linear models and permits identification without requiring all coefficients to be non-negative.
>
> We acknowledge that this point was not clearly explained and will revise the manuscript to clarify it.
>
> ---
>
> **Q3: Relationship between Assumptions 3.2 and 3.3**
>
> **A3:**  Thank you for this insightful question. Assumption 3.3 pertains solely to the outcome variable $Y$ (denoted as $W_{p+1}$), whereas Assumption 3.2 pertains to the treatment variables $W_2, \ldots, W_p$. Since they apply to different variable sets, Assumption 3.3 does not imply Assumption 3.2. We will make this distinction clearer in the revised text.
>
> ---
>
> **Q4: Required Assumptions for Theorem 3.6**
>
> **A4:**  Thank you for this careful and insightful observation.  You are correct—our intention was to show that the identification expression of postNDE, postNIE, and postTCE in the *first part* of Theorem 3.6 relies only on Assumptions 3.1 and 3.2. However, identification of the counterfactual term $\mathbb{E}(Y_{x_k^\ast, d_k^\ast} \mid x, \mathcal{E})$ in the *second part* of the theorem does require Assumption 3.3 via Lemma 3.4.
>
> We appreciate your comments and will revise the statement of Theorem 3.6 to explicitly include Assumption 3.3.
>
> ---
>
> **Q5: Clarification on Lemma 4.2**
>
> **A5:**  Thank you for your thoughtful question. The properties of the function $\rho_{x\to x'}(\cdot;y)$ serve important theoretical roles:
>
> - **Continuous differentiability** ensures that gradient-based optimization is feasible and reliable.
> - **Weak convexity** guarantees the existence of at least one global minimizer.
> - **Strict convexity** (within the interior of the support $S_{Y_{x'}}$  guarantees uniqueness of the minimizer, which ensures that the counterfactual mapping $\phi_{x \to x'}(y)$ is well-defined and stable.
>
> Together, these properties underpin the theoretical soundness of our estimation method and justify the nonparametric approach adopted. We will elaborate on this in the revised manuscript.
>
> ---
>
> **Q6: On the Support of the Conditioning Set**
>
> **A6:**  Thank you for the helpful comment!  Yes, we require $\mathrm{pr}(X = x, \mathcal{E}) \neq 0$ to ensure that the conditional distributions are well-defined. We have made this assumption explicit in the revised manuscript for clarity.
>
> ---

---

### Official Review · Reviewer_mnpa · 2025-03-10

**Overall Recommendation:** 4

**Summary:**

This paper focuses on the causal attribution analysis, which answers retrospective questions like "given that an outcome has occurred, how can we figure out how much each potential cause contributed to it?" Most existing literature on this are introduced with binary outcome variables, which is not the case in real-world analysis.

Therefore, in this paper, authors define a new set of "posterior causal estimands" for continuous outcomes. These include: posterior total causal effect, posterior natural direct effect, posterior natural indirect effect, and posterior intervention causal effect. The authors develop identifiability of these measures under certain assumptions. Further, for estimation of these quantities, a two-step approach is proposed.

---

## update after rebuttal:

I thank the authors for the response. I keep my score of acceptance.

**Claims And Evidence:**

Yes.

**Essential References Not Discussed:**

/

**Experimental Designs Or Analyses:**

Yes.

**Methods And Evaluation Criteria:**

Yes.

**Other Comments Or Suggestions:**

/

**Other Strengths And Weaknesses:**

Strengths:

1. The problem studied is necessary and crucial. It addresses a clear gap by extending causal attribution analysis to continuous outcomes, which is quite novel.

2. The technical development is well-grounded and rigorous. The paper is also well structured, with formulations of quantities built first, and then the identifiability guarantees under certain assumptions.

3. The real-world examples on hypertension and abnormal weights are interesting. One of my questions is: when only observational data is available, do we have any metrics to evaluate the quality of the estimated quantities, in terms of accuracy to the true quantities from counterfactual experiments?


Weaknesses:

1. The assumptions are strong, especially for the Assumption 3.2 (Monotonicity) so that there are "no prevention" relations. Are these assumptions in any way testable?

2. Except for listing such four posterior causal effect estimands, a discussion and practical guide on which to use under different scenarios is expected. On top of that, an analysis to showcase the connections between them, and also to their counterparts in the binary outcome case, are expected.

**Questions For Authors:**

/

**Relation To Broader Scientific Literature:**

/

**Theoretical Claims:**

Yes. I read the measures' formulations, the assumptions, and the theoretical claims about identifiability. I skimmed through the proofs. The results seem correct to me but I cannot guarantee.

---

> ### Author Rebuttal · Authors · 2025-03-29
>
> We sincerely thank the reviewer for the constructive feedback and the positive recommendation. We appreciate your recognition of our contribution and your valuable suggestions, which we will address in the revised manuscript.
>
> ---
>
>  **Q1.** *The assumptions are strong, especially for Assumption 3.2 (Monotonicity), so that there are “no prevention” relations. Are these assumptions in any way testable?*
>
> **A1.** Thank you for this insightful question. We agree that Assumption 3.2 is strong and may not always hold in practice. As discussed in Section 3.2 of the manuscript, although this assumption is not confirmed in the statistical sense—that is, it cannot be verified solely from observed data—it can be rejected under Assumption 3.1 (sequential ignorability), by checking whether certain inequality constraints on observed conditional probabilities hold in the data.
>
> To illustrate, consider the case with two binary treatment variables $X_1$ and $X_2$. The monotonicity assumption implies that if $X_1 = 1$, then $X_2$ must be 1; whereas if $X_1 = 0$, then $X_2$ can be either 0 or 1. This leads to the testable implication:
>  $\mathrm{pr}(X_2 = 1 \mid X_1 = 0) \leq \mathrm{pr}(X_2 = 1 \mid X_1 = 1).$
> If we observe in the data that $\mathrm{pr}(X_2 = 1 \mid X_1 = 0) = 0.7$ and $\mathrm{pr}(X_2 = 1 \mid X_1 = 1) = 0.5$, this inequality is violated, providing empirical evidence against monotonicity.
>
> We will incorporate these testable implications and further concrete illustrations into the revised manuscript.
>
> ---
>
> **Q2.** *Except for listing the four posterior causal effect estimands, a discussion and practical guide on which to use under different scenarios is expected.*
>
> **A2.** Thank you for the helpful suggestion. To clarify the roles of the proposed posterior causal effect estimands, we have added a summary table in the revised manuscript. A simplified version is provided below:
>
> | Estimand | Definition | Interpretation |
> |----------|------------|----------------|
> | **PostTCE** | $E(Y_{X_k=1} - Y_{X_k=0} \mid x, \mathcal{E})$ | Total effect of switching $X_k$ from 0 to 1, given treatments $x$ and observed event $\mathcal{E}$. |
> | **PostNDE** | $E(Y_{X_k=1, D_k(a_k, 0)} - Y_{X_k=0} \mid x, \mathcal{E})$ | Direct effect of changing $X_k$ while holding mediator fixed at its value under the reference value $X_k = 0$. |
> | **PostNIE** | $E(Y_{X_k=1} - Y_{X_k=1, D_k(a_k, 0)} \mid x, \mathcal{E})$ | Indirect effect  induced by changes in the intermediate variables $D_k$. |
> | **PostICE** | $E(Y_{X=x'} - Y \mid x, \mathcal{E})$ | Effect of changing the entire treatment vector from $x$ to $x'$. |
> | **ITE** | $Y_{x'} - Y_{x^*}$ | Individual-level contrast between two treatment configurations. |
>
> We have also revised the illustrative example. Suppose an individual has exposure profile $x = (\text{E}, \text{D}, \text{Hb}, \text{HD}, \text{CP})$ and observed outcome $Y > 140$. Given the observed evidence, the posterior estimands enable retrospective attribution in the following way:
>
> - PostTCE assesses the overall effect of a single factor (e.g., lack of exercise) on blood pressure.
> - PostNDE identifies the portion of the effect that is direct.
> - PostNIE captures the indirect path through variables like heart disease.
> - PostICE quantifies the joint impact of multiple exposures (e.g., poor diet and no exercise).
> - ITE compares the individual's outcome under two hypothetical exposure profiles.
>
> We will include this practical guidance in the revised manuscript to assist interpretation and application.
>
> ---
>
> **Q3.** *An analysis to show the connections between the estimands, and their counterparts in the binary outcome case, is expected.*
>
> **A3.** Thank you for your helpful comments. In the revised manuscript, we provide a unified set of definitions for the five posterior causal effect estimands under both continuous and binary outcomes, and clarify their applicability and identifiability.
>
> Specifically, let $ Y^*$ $ denote the binary outcome. Following the framework of Lu et al. (2023), we define the binary versions of the posterior causal estimands as follows:
>
> - PostTCE:  $\operatorname{postTCE}^*(X_k \Rightarrow\mathcal{E} \mid x,Y^* = 1) = E(Y^*_{X_k=1} - Y^*_{X_k=0} \mid x, Y^* = 1)$
> - PostNDE:  $\operatorname{postNDE}^*(X_k \Rightarrow\mathcal{E} \mid x,Y^* = 1) = E(Y^*_{X_k=1, D_k(a_k, 0)} - Y^*_{X_k=0} \mid x, Y^* = 1) $
> - PostNIE:  $\operatorname{postNIE}^*(X_k \Rightarrow \mathcal{E} \mid x, Y^* = 1) = E(Y^*_{X_k=1} - Y^*_{X_k=1, D_k(a_k, 0)} \mid x, Y^* = 1) $
> - PostICE: $\operatorname{postICE}^*(Y^*_{x'} \mid x, Y^* = 1) = E(Y^*_{x'} - Y^* \mid x, Y^* = 1) $
>
> While the definitions of $\operatorname{postNDE}^*$ and $\operatorname{postNIE}^*$ are conceptually consistent with their continuous-outcome counterparts, their identifiability under the binary setting has not yet been fully established in the literature.
>
> We will provide a unified presentation and discuss this point in the revised manuscript.
>
> ---

---

### Official Review · Reviewer_HUHT · 2025-03-12

**Overall Recommendation:** 4

**Summary:**

This paper addresses the causal attribution problem for continuous outcome variables, an interesting and realistic scenario compared to binary outcomes. It introduces a set of posterior causal estimands to retrospectively analyze causal attribution:  PostTCE, PostNDE, PostNIE, PostICE. Under assumptions of sequential ignorability, monotonicity, and perfect positive rank, the authors demonstrate the identifiability of these estimands, and propose an efficient two-step estimation method based on quantile matching. The theories are validated through hypertention dataset and toxicity risk dataset.

**Claims And Evidence:**

The claims regarding the identifiability of posterior causal estimands seem to be sound under stated assumptions.

**Essential References Not Discussed:**

There are related work discussing causal attribution problems in the context of Directed Acyclic Graphs (DAG) [1] and using Shapley Values [2].  It might be interesting to discuss the relation between them and posterior estimands.

[1] Schamberg, Gabriel, William Chapman, Shang-Ping Xie, and Todd P. Coleman. "Direct and indirect effects—An information theoretic perspective." *Entropy* 22, no. 8 (2020): 854.

[2] Jung, Yonghan, Shiva Kasiviswanathan, Jin Tian, Dominik Janzing, Patrick Blöbaum, and Elias Bareinboim. "On measuring causal contributions via do-interventions." In *International Conference on Machine Learning*, pp. 10476-10501. PMLR, 2022.

**Experimental Designs Or Analyses:**

Yes, I checked the experiments designs. It adequately satisfied the assumptions and the analysis is valid,.

**Methods And Evaluation Criteria:**

The proposed evaluation methods and datasets are relevant and standard for the issue being discussed.

**Other Comments Or Suggestions:**

Typo: L106 on the right column: PostDNE -> PostNDE

**Other Strengths And Weaknesses:**

Overall, I think the paper addresses an important and underexplored issue in causal inference for continuous outcomes. There are certain parts in the proof that I didn't follow because posterio causal estimands is not my expertise, and the proofs are largely built upon [Lu 2023] and [Li 2023] and quite dense, but based on my current understanding, the claims seem to be sound.

**Questions For Authors:**

Given that the assumptions are similar to to that of [Lu 2023] and [Li 2023] for binary outcomes, (i) ihow likely are assumptions 3.2 and 3.3 going to be satisfied in general for continuous outcomes? My worry is that the assumptions would be too strong. (ii) how difficult is it to be justified/tested out?

**Relation To Broader Scientific Literature:**

This paper mainly focus on extending attribution analysis from binary outcomes to continuous outcomes. It builds upon the line of work by Pearl (2000), Dawid et al. (2014), and especially recent studies by Lu et al. (2023) and Li et al. (2023).

**Theoretical Claims:**

I mainly checked the proofs of Theorem 3.6, but not for other corollaries. The part that I read is correct.

---

> ### Author Rebuttal · Authors · 2025-03-29
>
> We sincerely thank the reviewer for the positive evaluation and constructive suggestions, which have been very helpful in improving the clarity and quality of our manuscript.
>
> ---
>   **Q1.** Discuss relation to essential references.
>
> **A1.** Thank you for pointing out the related work by Schamberg et al. (2020) and Jung et al. (2022). Both are highly relevant to our goal of understanding and quantifying causal contributions, and—like our study—they offer principled frameworks for attributing the effects of multiple variables. While we share a common objective, our approach differs in several important ways.
>
> First, our method adopts a retrospective **causal attribution perspective**, conditioning on the observed outcome $Y$ to assess how each treatment component contributed to that specific realization. In contrast, Schamberg et al. and Jung et al. take an **effects-of-causes** perspective, focusing on how interventions influence the distribution of outcomes. Second, we introduce posterior causal estimands, defined based on observed treatment–outcome pairs, which allow for individual-level attribution. In comparison, Schamberg et al. rely on population-level, information-theoretic measures, while Jung et al. use do-Shapley values to quantify feature contributions under hypothetical interventions.
>
> We will cite both papers and discuss these connections in the revised manuscript.
>
> ---
>   **Q2.** *How likely are Assumptions 3.2 and 3.3 to be satisfied in general for continuous outcomes?*
>
> **A2.** Thank you for raising this interesting question. We believe that in many practical applications with temporal structures, progressive interventions, or latent rank-based heterogeneity, Assumptions 3.2 and 3.3 are generally plausible, or at least serve as reasonable modeling approximations.
>
> - **Assumption 3.2 (Monotonicity)** concerns the relationship among the treatment variables $X_1, \ldots, X_p$, rather than between treatments and the continuous outcome. Specifically, it assumes that earlier treatment components have non-negative effects on subsequent ones. This assumption is likely to be satisfied in sequentially administered or progressive treatments, where prior treatments increase the chance or intensity of receiving later ones. For example, in multi-stage educational or medical programs, earlier stages (e.g., early screening or primary education) often facilitate participation in later ones (e.g., advanced therapies or higher education).
>
> - **Assumption 3.3 (Perfect Positive Rank)** is more abstract but remains plausible in many applications. It assumes that the individual-level outcome $Y$ is determined by a stable, unobserved factor in a strictly increasing way. This holds approximately when treatment primarily shifts the overall level of the outcome without changing individuals' relative ranking. For example,  if a person has the lowest blood pressure under treatment, he is also expected to have the lowest blood pressure without treatment.  Such rank-preserving behavior is commonly assumed in structural equation models, linear models, and additive models such as
>
>   $$
>   Y = \alpha_0 + \alpha_1 X_1 + \cdots + \alpha_p X_p + \epsilon, \quad \text{or} \quad Y = f(X_1, \ldots, X_p) + \epsilon,
>   $$
>   where $\epsilon$ captures unobserved heterogeneity.
>
>
> We will incorporate the above discussions and examples in the revised manuscript.
>
> ---
>  **Q3.** *How difficult is it to justify or test these assumptions?*
>
> **A3.** Thank you for raising this insightful point. While Assumptions 3.2 and 3.3 are not statistically testable in the strict sense, their plausibility can be assessed through empirical strategies and domain expertise.
>
> - **Assumption 3.2** is **falsifiable** under Assumption 3.1 (sequential ignorability), by verifying inequality constraints on observed conditional probabilities. For example, in a setting with binary treatment variables $X_1$ and $X_2$, monotonicity implies: $
>   \operatorname{pr}(X_2 = 1\mid X_1 = 0) \leq \operatorname{pr}(X_2 = 1\mid X_1 = 1). $
> If we observe in the data that: $
>   \operatorname{pr}(X_2 = 1\mid X_1 = 0) = 0.7, \operatorname{pr}(X_2 = 1\mid X_1 = 1) = 0.5, $
>   then monotonicity is violated, offering empirical evidence against the assumption.
>
> - For **Assumption 3.3**, we first suggest assessing, in real data, whether a common unobserved rank variable exists across different treatment conditions. This can serve as an empirical check of the assumption’s plausibility. In addition, one may fit linear or additive models, examine residual patterns, and evaluate model fit using criteria such as AIC, BIC, or R$^2$. If these models can explain a substantial proportion of the outcome variation, this would provide indirect support for the reasonableness of the assumption.
>
>
> We will incorporate these discussions in the revised manuscript.
>
> ---
>   **Q4.** *Line 106: “PostDNE” → “PostNDE”*
>
> **A4.** Thank you for this careful observation. We will correct it in the revised manuscript.
>
> ---

---

### Official Review · Reviewer_31BM · 2025-03-13

**Overall Recommendation:** 4

**Summary:**

The submission describe a method for counterfactual analysis for continuous outcomes.
Identifiability conditions and results are given. Moreover based on minimization of a certain loss the authors propose an estimator and derive some theoretical properties.
A practical example using simulated data show the results of the proposed methods, moreover a real problem is developed.

**Claims And Evidence:**

Yes clear

**Essential References Not Discussed:**

No

**Experimental Designs Or Analyses:**

The experimental design and analysis seems sound and correct.
It mainly follows the experiments of Lu et al. 2023

**Methods And Evaluation Criteria:**

Yes they make sense

**Other Comments Or Suggestions:**

None

**Other Strengths And Weaknesses:**

The paper is well written and clear, it can be followed well.
The originality is somehow restricted to the extension of a previous approach to continuous outcomes.

**Questions For Authors:**

No

**Relation To Broader Scientific Literature:**

The submission extend the work of Lu et al. To the case of continuous outcomes..
The setting, definition and experiments are mainly taken from Lu et. al

**Theoretical Claims:**

I did not check proofs in the supplementary, the theoretical claims seems sounds.

---

> ### Author Rebuttal · Authors · 2025-03-28
>
> ---
>
> **Q1.** *The paper is well written and clear; it can be followed well. The originality is somehow restricted to the extension of a previous approach to continuous outcomes.*
>
> **A1.** We sincerely thank the reviewer for the positive and encouraging feedback. We truly appreciate your kind recognition of the manuscript’s clarity, the soundness of the proposed methodology, and the overall quality of both the theoretical and empirical components.
>
> While our work builds upon the framework introduced by Lu et al. (2023), our aim is not simply to apply it to continuous outcomes, but to extend it in several meaningful directions. In particular, we
> - formulate a new class of posterior causal attribution estimands suitable for continuous responses,
> - establish their identification under structured assumptions tailored to the continuous setting, and
> - design a two-step estimation strategy that facilitates practical implementation and retrospective interpretation.
>
> We are grateful for your comments and will revise the manuscript to better highlight these contributions and more clearly position our work in relation to the existing literature.
>
> ---

---

> > ### Comment · Reviewer_31BM · 2025-04-01
> >
> > I thank the authors for the responses.
> > I will keep my positive score

---

> > > ### Author Response · Authors · 2025-04-03
> > >
> > > We sincerely thank the reviewer for the positive evaluation and continued support of our work.

---

### Decision · Program_Chairs · 2025-05-01

**Decision:**

Accept (spotlight poster)

**Comment:**

The paper studies the problem of causal attribution with continuous outcome variables. The authors introduce several posterior causal estimands and present conditions for identifiability, along with an effective estimation procedure. The problem addressed is novel, and the analysis is built on a solid theoretical foundation.